# Near-Certain Reasoning: Bridging the Formalization Gap Between Language Models and Logical Solvers

## Abstract

Robustness of reasoning remains a challenging problem for large language models, and addressing it is crucial for advancing the reliability and practical application of AI-driven reasoning systems. We introduce Semantic Self-Verification (SSV), a novel approach that addresses the key challenge in combining language models with the rigor of logical solvers: to accurately formulate the reasoning problem from natural language to the formal language of the solver. SSV produces strong abstract formalizations of problems by verifying and refining them against concrete instantiations that are generated by the model and verified by the solver. In addition to significantly advancing the overall reasoning accuracy over the state-of-the-art, a key novelty that this approach presents is a feature of verification that has near-perfect precision over a significant coverage of cases, as we demonstrate on open reasoning benchmarks. We propose such *near-certain reasoning* as a new approach to reduce the need for manual verification in many cases, taking us closer to more dependable and autonomous AI reasoning systems.

## 1 Introduction

Logical reasoning remains an open challenge for large language models (LLMs). While such models have exhibited reasoning ability in various domains, the reasoning is often fragile and error-prone, especially as tasks get more complex. Many recent approaches have made notable advancements in this active area of research. Chain-of-thought (CoT) prompting has demonstrated how the quality of reasoning can be improved by prompting the model to explicitly generate the steps of reasoning in natural language before arriving at the final answer (Wei et al. (2022)). Variants of CoT and other related prompting and fine-tuning approaches have shown further improvements (Zhou et al. (2023); Wang et al. (2023); Yu et al. (2024); Weng et al. (2023); Creswell et al. (2023)). To address the logical inconsistencies that can arise in such natural language based approaches, another interesting direction is to incorporate LLMs with logical solvers or automated reasoning tools (Pan et al. (2023); Ye et al. (2023)). Rather than directly attempting reasoning with the LLM, these approaches use the LLM to infer a formal representation of the problem as a program that can be executed by the solver, as such automated reasoning tools guarantee logically sound inference by construction.

While these approaches have demonstrated relative improvements in accuracy, we are still far from achieving robustness and reliability of reasoning. For instance, Figure 1a shows an example reasoning problem from the Law School Admissions Test on analytical reasoning (Zhong et al. (2022)). On tasks of such complexity, the best reported accuracy, achieved by a solver-augmented system, is only 43% (Pan et al. (2023)). Such lack of reliability especially hinders the practical usability of existing approaches: for example, if a system demonstrates 70% accuracy on benchmarks, then in practice the user can only be 70% confident that the answer is correct on an arbitrary new task. Hence the burden of verifying correctness is *always* on the user, which can be especially difficult and error-prone for complex reasoning tasks. Therefore, having a reliable signal of correctness with high confidence can be hugely beneficial to help reduce the overall manual effort and cost of verification.

In this work, we propose a new approach to addressing deductive reasoning problems called *Semantic Self-Verification* (SSV), which offers two key benefits: (1) it improves the overall accuracy of reasoning significantly over SoTA, and (2) it provides a novel feature of verification that

In a repair facility, there are exactly six technicians: Stacy, Urma, Wim, Xena, Yolanda, and Zane. Each technician repairs machines of at least one of the following three types—radios, televisions, and VCRs—and no other types. The following conditions apply: Xena and exactly three other technicians repair radios. Yolanda repairs both televisions and VCRs. Stacy does not repair any type of machine that Yolanda repairs. Zane repairs more types of machines than Yolanda repairs. Wim does not repair any type of machine that Stacy repairs. Urma repairs exactly two types of machines. Which one of the following pairs of technicians could repair all and only the same types of machines as each other?

**(A)** Stacy & Urma

**(B)** Urma & Yolanda

**(C)** Urma & Xena

**(D)** Wim & Xena

**(E)** Xena & Yolanda

```
technicians = [Stacy, Urma, Wim, Xena, Yolanda, Zane]
machines = [radios, televisions, VCRs]
repairs = Function('repairs', technicians_sort, machines_sort, BoolSort())

pre_conditions = []
pre_conditions.append(ForAll([t], Sum([If(repairs(t, m), 1, 0) for m in
machines]) >= 1))

# CONSTRAINT: Xena and exactly three other technicians repair radios.
pre_conditions.append(And(repairs(Xena, radios), Sum([If(And(t != Xena, repairs
(t, radios)), 1, 0) for t in technicians]) == 3))

# CONSTRAINT: Yolanda repairs both televisions and VCRs.
pre_conditions.append(And(repairs(Yolanda, televisions), repairs(Yolanda,
VCRs)))

# CONSTRAINT: Stacy does not repair any type of machine that Yolanda repairs.
pre_conditions.append(ForAll([m], Not(And(repairs(Stacy, m), repairs(Yolanda,
m)))))

# CONSTRAINT: Zane repairs more types of machines than Yolanda repairs.
pre_conditions.append(Sum([If(repairs(Zane, m), 1, 0) for m in machines]) > Sum
([If(repairs(Yolanda, m), 1, 0) for m in machines]))

# CONSTRAINT: Wim does not repair any type of machine that Stacy repairs.
pre_conditions.append(ForAll([m], Implies(repairs(Stacy, m), Not(repairs(Wim,
m)))))

# CONSTRAINT: Urma repairs exactly two types of machines.
pre_conditions.append(Sum([If(repairs(Urma, m), 1, 0) for m in machines]) == 2)

# OPTION A:
if is_sat(And(ForAll([m], repairs(Stacy, m) == repairs(Urma, m)))): print('(A)')

# OPTIONS B to E stated similarly ...
```

(a) Sample reasoning problem   (b) Problem formalization as a Z3 solver program

Figure 1: Sample question from the Law School Admissions Test dataset on analytical reasoning tasks (AR-LSAT), and its formalization as code in the Z3 theorem prover language

has *near-perfect* precision. In our problem formulation, in addition to producing an answer to a given question, the system also indicates if it was able to *verify* the correctness of the answer: Question $\rightarrow$ (Answer, isVerified). This problem formulation is similar to confidence estimation in machine learning, where the system provides a score of confidence in addition to the answer. However, similar to selective classification (Chow (1970)), in our case the isVerified indicator is a boolean rather than continuous value: if true, it indicates a "near certain" confidence in the correctness of the answer, and otherwise there is no specific indication of confidence. The goal is to provide a high confidence verification mechanism that can be used to reduce the need for manual checking in the cases where verification succeeds.

At its core, our approach addresses the key challenge in combining large language models with the robust reasoning of logical solvers: the formulation of a problem from informal natural language (NL) to the formal representation that is a program executable by the solver. For example, Figure 1b shows the formal representation of the problem expressed in natural language in Figure 1a. In this case the formalization is expressed as code in the language of the Z3 SMT solver (de Moura & Bjørner (2008)), which is a state-of-the-art industrial strength theorem prover that can produce the correct answer when given these correctly-expressed formal constraints. The crucial task, therefore, is for the LLM to correctly translate the NL problem description to such a formal representation, and this is where language models can make significant errors, especially for tasks of such complexity.

Hence the main goal of the SSV approach is to verify that the formal representation is true to the original problem. This notion of verification is inspired by how humans often create formalizations of problems expressed in natural language. For instance, when school students are solving math word problems, they need to first create the right algebraic equation that represents the problem, before they can solve it to get the answer. To ensure that their translation to an abstract equation represents the problem correctly, they are encouraged to consider various concrete instances of the problem and to check that the abstract equation consistently satisfies those instances so that it all "makes sense". In the same way, in the SSV approach, rather than just doing a single abstract translation from NL to a formal representation, we also use the LLM to additionally generate various *concrete instantiations*, or examples, of the general constraint, which are used as test cases to check the correctness of the abstract formalization. Using the logical solver, we verify that each of these

instantiations is consistently satisfied by the formal representation. If all of these distinct semantic relationships consistently hold, then verification passes.

We note that any notion of verification from natural to formal language cannot provide formal correctness guarantees, since natural language itself is inherently informal and often ambiguous. However, as we demonstrate empirically, a passing verification in our case indicates a *near certain* confidence in the answer correctness since multiple independent semantic relationships are consistently satisfied. In this respect, our approach is akin to a consensus-based ensemble as it is based on agreement between multiple independent predictors (Zhou (2012)). However, rather than all predictors addressing *the same* task, we have a *semantic ensemble* of predictors that are addressing different but semantically related tasks (making abstract and concrete inferences) and the logical solver verifies the formal consistency between these. We also note that unlike standard proposer-verifier approaches, in our case there does not exist a verifier that can check correctness of a proposed solution (a formalization). Thus our proposer model proposes both a solution and the test cases and the verifier can only check *consistency* between these rather than *correctness* of the solution.

Moreover, having such a high precision verification mechanism also allows us to improve the formalization itself, in two different respects. Firstly, any failing instantiation can be used as concrete guidance to refine the formalization further, as it can hint at potential errors in the formalization. This is similar to error-based refinement in code generation techniques (Chen et al. (2024)), except that here we are guided by *semantic* errors inferred from the instantiations rather than just *syntactic* execution errors in the code. Secondly, given a high precision verifier, we can also explore the search space more extensively until we find a formalization that can pass verification. We show how creating multiple candidate formalizations at different LLM temperatures and choosing the ones that pass our verification yields a higher overall accuracy.

Our evaluation demonstrates how the SSV approach achieves a significant increase in overall accuracy, as well as a near-perfect precision (or selective accuracy) on the verified cases. Figure 2 highlights the results for the most challenging AR-LSAT law school tests dataset. Though better than direct LLM inference and CoT, the accuracy of the best performing existing system (the solver-augmented LOGIC-LM approach by Pan et al. (2023)) is at 43%, while SSV achieves a significantly higher accuracy of 71.3%, which also surpasses the average human performance. Moreover, the precision (or selective accuracy) of the 21.7% of cases that it is able to verify is 100%. This means that a 21.7% reduction in manual verification effort can potentially be made on tasks of such high complexity. In our full evaluation we also show higher accuracy and coverage of verified cases on other open reasoning datasets.

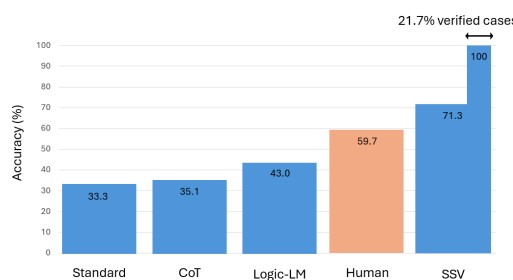

Figure 2: Towards near-perfect reasoning: SSV achieves new SoTA accuracy and 100% verification precision on the AR-LSAT law school tests dataset (*all systems using GPT-4 as base LLM*).

In summary, we make the following contributions in this work: (1) We propose the problem formulation of returning a boolean high-confidence verification indication in addition to the answer, which can be used to reduce manual cost of verification. (2) We present the novel technique of semantic self-verification, which uses concrete instantiations to verify the correctness of the problem formalization. (3) We show how SSV can also improve the formalization itself through instantiation-guided refinement and exploration of multiple candidate formalizations. (4) We present an extensive evaluation on five open benchmarks that shows a significant increase in overall accuracy over SoTA, as well as near-perfect selective accuracy over a significant coverage of verified cases.

## 2 INFERRING THE RIGHT FORMALIZATION: A MOTIVATING EXAMPLE

Let us consider the third constraint from the technicians problem in Figure 1b, which requires that "Stacy does not repair any type of machine that Yolanda repairs". Figure 3 illustrates how the SSV approach works in this case. A direct translation using the LLM may produce an incorrect abstract formalization of this constraint as shown in Figure 3a, where the constraint is asserted only *for some*

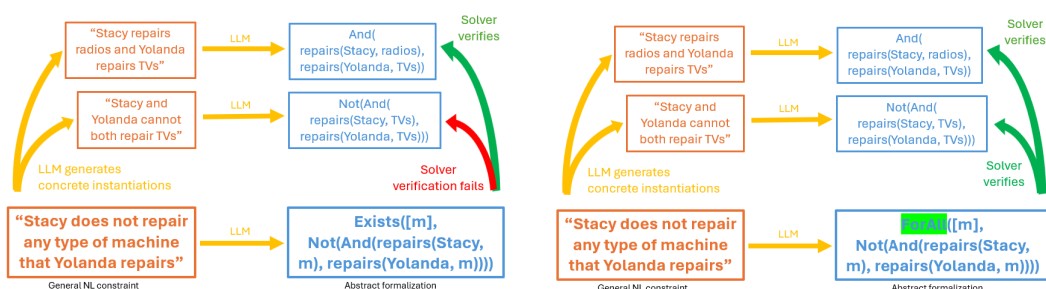

(a) Incorrect formalization (uses Exists quantifier)   (b) Correct formalization (uses ForAll quantifier)

Figure 3: Semantic self-verification of a general constraint: one concrete instantiation fails for the wrong formalization in (a), while both instantiations are verified for the correct formalization in (b)

machine rather than *for all* machines because the Exists quantifier is incorrectly used. However, in the SSV approach, we use the LLM to also infer simple concrete instantiations, or examples, of the general NL constraint. For instance, a concrete positive example is that Stacy repairs radios and Yolanda repairs TVs. A concrete negative example is that Stacy and Yolanda cannot both repair TVs. After inferring these examples in NL, we also use the LLM to translate them to formal expressions in the language of the solver. We then use the logical solver to check that each of these expressions is satisfiable under the abstract formalization. In Figure 3a we see that the second instantiation fails verification because the abstract formalization does not assert the condition for all machine types, so it still allows for the possibility that Stacy and Yolanda can both repair TVs.

However, with the correct formalization in Figure 3b that uses the ForAll quantifier, we see that both instantiations pass the solver verification, since the abstract formalization correctly disallows that *any* machine can be repaired by both Stacy and Yolanda. In the same way, SSV verifies all of the constraints identified in the full program by inferring concrete instantiations for them using the LLM. For instance, for the first constraint in Figure 1b it may infer a positive example that Xena, Urma, Wim and Stacy repair radios, and a negative example that only Xena and Urma repair radios.

## 3 SEMANTIC SELF-VERIFICATION

In this section we describe the semantic self-verification approach for solving reasoning problems, which is based on generating programs that are verified and refined by concrete instantiations. The main algorithm is shown in Figure 4, which shows the top-level flow and key components of the approach. As discussed before, in our problem formulation, the algorithm takes as input a question (Q), such as the technicians problem from Figure 1a, and outputs a pair of values which are the answer to the question and a boolean flag that indicates if verification has succeeded. Figure 4 also shows the other configuration parameters that the algorithm takes: the particular LLM and solver to use, as well as the temperature values for the LLM to explore and the maximum number of repair attempts. We shall first describe the general algorithm outline and then discuss the key phases in more detail.

**Require:** Q        // *the question*
**Require:** LLM       // *the language model*
**Require:** Solver       // *the logical solver*
**Require:** Temperatures    // *LLM temperatures to try*
**Require:** MaxRepairs    // *maximum repair attempts*
1: $A_{\text{best}} \leftarrow \varnothing$
2: **for each** $T \in$ Temperatures **do**
3:   $P \leftarrow$ **GenProgram**(LLM, $T$, Solver, Q)
4:   **while** $P \neq \varnothing$ **and under** MaxRepairs **do**
5:     $A \leftarrow$ **ExecuteProgram**(Solver, $P$)
6:     **if** $A_{\text{best}} = \varnothing$ **then**
7:       $A_{\text{best}} \leftarrow A$
8:     $\mathcal{I} \leftarrow$ **GenInstantiations**(LLM, $T$, $P$)
9:     $I_{\text{fail}} \leftarrow$ **Verify**(Solver, $\mathcal{I}$, $P$)
10:     **if** $I_{\text{fail}} = \varnothing$ **and IsWellFormed**($P$) **then**
11:       **return** $(A, \text{True})$
12:     $P \leftarrow$ **RepairProgram**(LLM, $T$, Q, $P$, $I_{\text{fail}}$)
13: **if** $A_{\text{best}} = \varnothing$ **then**
14:   $A_{\text{best}} \leftarrow$ **InferLLMAnswer**(LLM, Q)
15: **return** $(A_{\text{best}}, \text{False})$

Figure 4: The Semantic Self-Verification Algorithm

For each temperature value to be explored, the algorithm begins by using the LLM to infer a program $P$ that can be executed by the solver to answer the question Q, such as the program from Figure 1b for the technicians problem. If an executable program is successfully generated ($P \neq \varnothing$), then we enter the verification loop (line 4). Here, we first execute the program using the solver to obtain an answer. Then to perform verification, we first infer concrete instantiations $\mathcal{I}$, which are test cases for each of the different constraints and options that the program $P$ contains, such as the six constraints and five options in the technicians program from Figure 1b. We attempt to verify that each of these instantiations is formally satisfiable using the solver and return any failing instantiation $I_{\text{fail}}$. For example, for the third constraint in the technicians program, we may infer the instantiations as in Figure 3a and obtain the failing instantiation "Stacy and Yolanda cannot both repair TVs". If there is no failing instantiation found (which would be the case in Figure 3b) and the program $P$ also satisfies some general well-formedness properties, then we return its answer $A$ along with verification success (line 11).

If verification fails, then we attempt to repair the program $P$ using the LLM and any failing instantiation found, as this instantiation can provide information about why the constraint that it instantiates may be implemented incorrectly. For example, the failing instantiation from Figure 3a may guide the LLM to infer that the condition should be asserted for all machine types using the forall quantifier as shown in Figure 3b. After obtaining the repaired program, we repeat the verification loop. If none of the answers could be verified at any of the temperatures in any repair attempts, then we exit the outer loop at line 13. If no answer was found at all so far (i.e. no executable program could be inferred), then we fall back to an answer by direct inference using the LLM with a chain-of-thought prompt, as done in prior work (Pan et al. (2023)). We then return the best answer along with verification failure. We next discuss some of the key phases of the algorithm in more detail.

**Program generation.** The GenProgram function in Figure 4 uses the LLM to generate a program that can be executed by the solver to address the given problem. A basic implementation of this could just be to use a direct LLM prompt to generate the solver code. However, we also utilize some effective techniques from the code generation literature to optimize the code quality. Firstly, we use error-based refinement, where if the generated program produces any syntax or execution errors then these are fed back to the LLM to repair the errors and obtain an executable program. This is a common approach to code generation with LLMs (Chen et al. (2024)), and has also been applied to reasoning domains (Pan et al. (2023)). Secondly, when direct code-generation fails to produce executable code, we also attempt a compositional approach (Khot et al. (2023); Pourreza & Rafiei (2024)), where the program is generated incrementally for each of the constraints identified from the original problem. Such approaches provide for better code generation to obtain executable code as compared to direct LLM prompting alone, which can produce code with syntax errors, etc. Our compositional code generation and refinement prompts are shown in Appendix A.1.

**Semantic verification.** While the above code generation approaches help to obtain an executable solver program, they do not address any *semantic* issues that may be present in the program: whether it accurately implements the intended constraints from the original problem. This is the main issue that SSV addresses by first generating concrete instantiations of the various constraints specified in the problem and then verifying that these instantiations are satisfied by the generated program. The GenInstantiations function first parses the generated program $P$ to extract each of the constraints as well as their NL descriptions. Our program generation phase creates programs with an explicit structure $P_{init} + C_1... + C_N + O_1 + ...O_M$, where there is an initial definitions segment $P_{init}$, followed by the constraints and options that are demarcated in explicit segments along with their NL descriptions stated as comments (e.g. "#CONSTRAINT:" and "#OPTION:" comments in Figure 1b). This structure is utilized to parse the constraints along with their respective NL descriptions from the program. For each NL description of a constraint, we use the LLM to infer concrete instantiations for it. Although in general we can generate an arbitrary number of instantiations for a given constraint, in our particular implementation prompt we ask the LLM to generate one positive example (where the constraint is satisfied) and one negative example (where the constraint is violated). Each of these examples is also translated as expressions in the language of the solver (as shown in Figure 3). The prompt for generating instantiations is shown in Appendix A.2.

Once we obtain the list of all instantiations $\mathcal{I}$, we next verify if each of them is consistent with its respective constraint. For each instantiation, given the initial definitions code segment of the program $P_{init}$, the constraint code $C$, and the instantiation expression $I$, the Verify function creates and executes a solver program $P_{init} + C + I$ that checks if the combination of the constraint

and instantiation is logically satisfiable. If verification fails, it returns the first failing instantiation $I_{\text{fail}} \in \mathcal{I}$. Apart from checking the concrete instantiations, we also check some general logical well-formedness properties of the program (`IsWellFormed` function). These include (1) structural checks to ensure the program is generated according to the format described above, (2) that the program returns some answer and does not return multiple answers, and (3) checks for degenerate expressions in the program that are logical tautologies or vacuously true implications, which tend to be redundancies or over-simplifications in the problem formalization.

**Semantic program repair.** If semantic verification fails and we have found a failing instantiation $I_{\text{fail}}$, the `RepairProgram` function uses the failing instantiation to attempt to repair the original program $P$ using the LLM if no answer has yet been found. This is similar to error-based program repair with LLMs, except that in this case it is a *semantic* repair based on the instantiation inferred by the LLM itself, rather than a syntactic or execution error in the program. In our repair prompt, we provide the initial definitions code, the constraint code and its NL description, and the instantiation expression that failed verification. We prompt the LLM to first analyse if the error is in the initial definitions, the constraint code or the instantiation itself (in a chain of thought fashion) and then to infer the corrected code. The prompt used for semantic program repair is shown in Appendix A.3.

## 4 EVALUATION

We present an evaluation of our SSV technique on existing open benchmarks for logical reasoning. The main goal of our evaluation is to determine the effectiveness of SSV with respect to two key aspects: (1) Improving the general accuracy of reasoning over existing baselines and (2) Providing a high quality verification mechanism: the correctness of verification (precision) and how many cases can be verified (coverage).

*Datasets.* We use five common datasets for evaluating logical reasoning tasks. To help in a direct comparison with the relevant baselines, we use the same datasets that were used in Pan et al. (2023). All datasets exist in a standard multiple-choice format, where each task comprises of a problem statement, a question, and potential answer options, as in the example shown in Figure 1a.

**PrOntoQA** is a dataset of synthetic deductive reasoning tasks for testing LLMs (Saparov & He (2023)). We use the most challenging version of the fictional characters dataset as identified in that work, which are tasks requiring 5 hops of reasoning. This is a total of 500 tasks in the test set with 2 answer options (True or False). **ProofWriter** is a widely used dataset for logical reasoning (Tafjord et al. (2021)) which, in contrast to PrOntoQA, has problems that are framed in a more naturalistic language. We use the open-world assumption subset with the most challenging tasks requiring 5 hops of reasoning. We use the same set used in Pan et al. (2023), where the test set contains 600 tasks that have 3 answer options (True, False or Unknown). **FOLIO** is an expert-crafted dataset designed for logical reasoning (Han et al. (2022)). The problems are closely aligned with real-world knowledge and are also phrased in highly natural language, requiring complex first-order logic reasoning for their solutions. We evaluate using the entire FOLIO test set, which contains 204 examples that have 3 answer options (True, False or Unknown). **LogDeduction** is another reasoning dataset from the BigBench collaborative benchmark (Srivastava et al. (2023)). The tasks mainly involve deducing the sequence order of objects based on a given set of arbitrary conditions. We evaluate using the complete test set, which consists of 300 tasks, each with 3,5 or 7 options for answers. **AR-LSAT** is a dataset that is created from a compilation of all analytical reasoning questions from the Law School Admission Test (LSAT) administered between 1991 and 2016 (Zhong et al. (2022)). This is a particularly challenging dataset, where even state-of-the-art models have only achieved performance that is a little better than random guessing (Pan et al. (2023); Liang et al. (2023)). The test set consists of 230 questions with each question having 5 possible answer options.

*Baselines.* We compare our technique against three baselines, which represent approaches of reasoning using the LLM alone, as well as the combination of formal logical solvers with LLMs. Each of these baselines and our own system is parametric in the LLM used, and in our experiments we investigate all systems with both the GPT-4 model (a current best general LLM for reasoning) as well as the weaker GPT-3.5 model from Open AI. We use the baselines and their results for these models as reported in Pan et al. (2023). The baselines are as follows.

| Dataset | General Accuracy | | | | SSV Verification | |
|---|---|---|---|---|---|---|
| | Standard | CoT | Logic-LM | SSV | Coverage | Precision |
| AR-LSAT | 33.3 | 35.1 | 43.0 | **71.3** | 21.7 | 94.0 (100.0) |
| FOLIO | 69.1 | 70.6 | 78.9 | **80.9** | 25.0 | 98.0 (100.0) |
| LogDeduction | 71.3 | 75.3 | 87.6 | **89.7** | 43.7 | 100.0 |
| PrOntoQA | 77.4 | 98.8 | 83.2 | **100.0** | 66.0 | 100.0 |
| ProofWriter | 52.7 | 68.1 | 79.7 | **98.0** | 75.2 | 98.7 (100.0) |

Figure 5: General accuracy of SSV approach and baselines, and the precision/coverage of SSV verification. Results shown are for GPT-4 used as the underlying model for all systems. *Precision values in brackets in green are the actual values in the corrected datasets.*

**Standard** is the direct approach of prompting the LLM, which leverages in-context learning to directly answer the question. **CoT** is the Chain-of-Thought technique (Wei et al. (2022)), which adopts a step-by-step problem-solving approach, using the LLM to first generate explanations before providing the final answer. **Logic-LM** is a state-of-the-art technique for combining LLMs with formal logical solvers to improve the robustness of reasoning (Pan et al. (2023)). The LLM is prompted to produce a representation of the problem as a formal solver program, which is then executed to produce the final answer. Finally, **SSV** is the implementation of our semantic self-verification technique, as shown in Figure 4. In our concrete implementation, we use the Z3 SMT solver as the logical solver (de Moura & Bjørner (2008)). The exact same prompts are used for both models, where 1-4 few shot examples were chosen from across the training datasets for each prompt (prompts shown in the Appendices). In the full implementation we set the SSV algorithm parameters MaxRepairs = 2 and Temperatures = $[0, 0.3, 0.4, 0.5]$ (exploring the lowest and mid-range temperatures), and also report on variations of these parameters in the ablation analysis.

### 4.1 RESULTS

**Main results**   The main results are shown in Figure 5, where all systems have been run with the GPT-4 model as the underlying LLM. The figure shows both the general accuracy of all systems as well as the precision and coverage of verification provided by our SSV technique. The general accuracy refers to the percentage of correct answers achieved by the system among all cases in the dataset. For SSV verification, the precision refers to the percentage of cases where the answer is correct among all cases which the SSV technique signalled as verified. The coverage refers to the percentage of cases that are signalled as verified by SSV among all cases in the dataset. We make the following key observations from these results:

1. SSV outperforms all baselines in terms of general accuracy. Our technique achieves a higher general accuracy over all baseline systems across all datasets. We especially note the drastic increase of 28.3% over the current best Logic-LM system on the most difficult AR-LSAT dataset. This shows the strong effectiveness of our technique in producing robust problem formalizations in contrast to just a direct LLM translation from the natural language description to the solver program.

2. SSV verification shows perfect empirical precision across all datasets. With the underlying GPT-4 model, we have found that the precision of verification with SSV is 100% on all of the datasets. Interestingly, on three of the datasets (AR-LSAT, FOLIO and ProofWriter), our verification mechanism actually discovered a few erroneous cases that we have checked were assigned wrong answers in the datasets. However, for consistent comparison to all baselines, in Figure 5 we have stated all numbers according to the original datasets (with the slightly lower precision values due to the incorrectly labelled cases). We provide explanations for the few correction cases in Appendix A.4 (for the AR-LSAT cases, we were able to also verify that our corrections are consistent with the original test question answers [1]). Such empirically perfect precision on these datasets demonstrates the very high level of confidence that SSV verification can provide for complex reasoning problems.

3. SSV verification shows significant coverage across all datasets. Although the precision is very high, we know that SSV verification does not always succeed. However, we find that the coverage

---

[1] https://img.cracklsat.net/lsat/pt/pt80.pdf

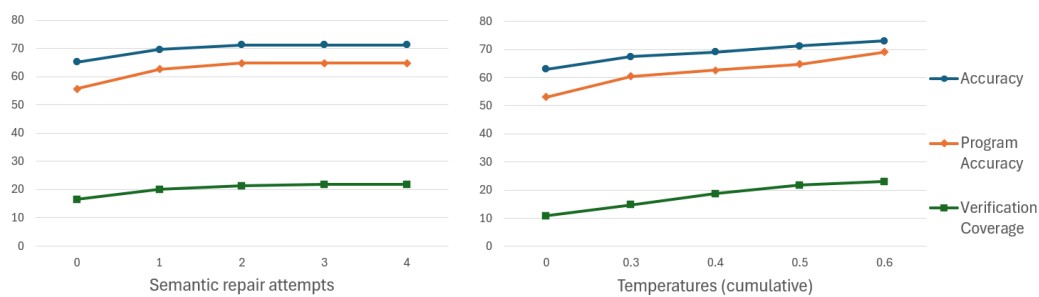

Figure 6: Semantic repair attempts and temperature variations on AR-LSAT

is significant across all datasets, with the lowest coverage of 21.7% on the most difficult AR-LSAT dataset. As expected, we find the coverage increases on the relatively easier datasets, with a verification coverage of up to 75.2% on ProofWriter. This significant coverage of verification shows that the SSV approach can help in avoiding manual human verification in a significant proportion of cases to reduce overall cost and effort.

**Effect of semantic repair and temperature exploration**   Figure 6 illustrates the effects of varying the number of semantic repair attempts (`MaxRepairs`) and temperatures (`Temperatures`) on the AR-LSAT dataset. We examined the effects on the three metrics of overall accuracy, the program accuracy (how often program generation was successful rather than fall-back to direct LLM answer), and the coverage of cases where verification succeeds.

In total there was a 6.1% gain in accuracy with semantic repairs, and 10.0% with all temperature explorations. The total gain in verification coverage was 5.2% for repair and more than doubled for temperature explorations, with a gain of 12.2% over the initial 10.9% coverage. In general, we found that for both repair attempts and temperature explorations, the gains were initially higher and then started to diminish, for both accuracy and verification coverage. For repair in particular, there was no improvement in any metric after 3 attempts, while temperature explorations continued to show some gains up to temperature 0.6. The gap between program accuracy and overall accuracy also reduced as repair attempts and temperature explorations increased (dropping from 9.8% to 5.2% on average), showing that program generation starts contributing more with these features.

| Dataset | General Accuracy | | | | SSV Verification | |
|---|---|---|---|---|---|---|
| | **Standard** | **CoT** | **Logic-LM** | **SSV** | **Coverage** | **Precision** |
| AR-LSAT | 20.3 | 17.3 | 26.4 | **28.3** | 0 | - |
| FOLIO | 45.1 | 57.4 | **62.7** | 59.3 | 1.5 | 100.0 |
| LogDeduction | 40.0 | 42.3 | **65.7** | 48.3 | 0 | - |
| PrOntoQA | 47.4 | 67.8 | 61.0 | **72.8** | 4.2 | 95.2 |
| ProofWriter | 35.5 | 49.2 | 58.3 | **72.5** | 16.2 | 94.8 (95.9) |

Figure 7: Results for GPT-3.5 model: general accuracy of all systems and SSV precision/coverage. *Precision values in brackets in green are the actual values in the corrected datasets.*

**Evaluation on GPT-3.5**   We also evaluated our system and all baselines using GPT-3.5 as the underlying LLM. The results are shown in Figure 7. Firstly, we note that while the general accuracy of all systems drops significantly with this weaker model, our SSV system still performs best overall, with an average accuracy of 56.2%. However, Logic-LM performs better than SSV on FOLIO and LogDeduction (this could be partly due to differences in the code generation quality for the different solver languages that Logic-LM uses for these datasets).

Secondly, we observe that while the coverage of SSV verification also drops significantly, with two of the more difficult datasets (AR-LSAT and LogDeduction) having no coverage at all, the precision of SSV is very minimally affected. On the datasets where there is coverage, we still see an average precision of 97%. This demonstrates an important property of reliability of SSV verification:

even for weaker models, if verification succeeds then it is still very reliable (and much more reliable than general accuracy), though it may succeed much less often. In practical terms, such reliability could even allow one to adopt a tiered strategy to optimize costs: trying weaker (cheaper) models for tasks first and fall-back on more expensive models if verification fails.

**Runtime performance**   The median runtime per task was 152 seconds (first quartile: 108s, third quartile: 267s) and mean 249s over a sample of 250 cases. More details and a discussion of potential optimizations to the SSV algorithm can be found in Appendix A.6.

## 5   LIMITATIONS AND FUTURE DIRECTIONS

Since natural language is informal and ambiguous, any verification approach with NL specifications cannot guarantee full correctness. Although SSV verification provides near-perfect empirical precision (100% with GPT-4), we discuss here the kinds of errors that are possible in SSV, which are illustrated by the few failing cases we found with GPT-3.5. In total, we found one case in PrOntoQA and four cases in ProofWriter that passed verification with an incorrect answer.

1. *Concrete instantiations are insufficient*. Since the approach is based on verification with respect to concrete examples (test cases), these may not test all aspects of the general constraint, especially all corner cases. Two questions failed for this reason with GPT-3.5. For example, in one case there were two separate conditions "Gary is nice" and "Gary is kind" in the original problem that were conflated to use the same predicate "is_kind(Gary)" in the formalization. If a concrete instantiation were generated that asserted "Gary is nice but not kind" then this would have detected the error.

2. *Concrete instantiation and program are both mutually consistent but wrong*. This is the unlikely case where both the program and the test case have the same error and therefore pass verification. We found only one such case which was a rather confusingly trivial error: for some reason the constraint "Fiona is quiet" was translated as its negation "Not(is_quiet(Fiona))" in both the program and the concrete instantiation independently generated by GPT-3.5.

3. *Missing or superfluous constraints*. In such cases the LLM may miss adding some constraints or add new constraints to the program that are not specified in the problem. Since our approach depends on the constraints being explicitly demarcated and parsed from the LLM-generated program, any errors by the LLM here can lead to potential failures in the verification. Two of the GPT-3.5 failure cases were caused by superfluous constraints being added. For example, in one case, the condition that was to be checked in the question was itself added as a constraint in the program.

In general, as we have found in our evaluation, such errors are rare and more likely in weaker LLMs, and can be expected to reduce further as LLMs mature. Errors such as (1) and (2) can also be reduced with a more exhaustive examples inference strategy, as in our implementation we took the simple approach of generating only 1 positive and 1 negative example per constraint. Class (3) errors stem from issues in the very basic structural consistency that is expected that the constraints expressed in the program match those from the original problem. While such basic consistency checks are less of an issue in mature LLMs such as GPT-4, one can also consider training simple specialized modules to check these core structural properties with high accuracy.

Another interesting direction is reasoning with missing background knowledge, which SSV does not handle as it focuses on pure deductive reasoning. Using LLMs to infer missing information before applying SSV can both enhance inference and also highlight missing assumptions to the user.

## 6   RELATED WORK

**Reasoning with LLMs**. Improving the robustness of reasoning in large language models is a very active area of research, and many recent approaches have made significant advancements. One direction of work has been to fine-tune or train specialized models that show improved reasoning ability (Tafjord et al. (2022); Clark et al. (2020); Yang et al. (2022)). Another direction has been to develop sophisticated prompting strategies to elicit better reasoning from LLMs. Chain-of-thought prompting (Wei et al. (2022)) has shown how the quality of reasoning can be improved by prompting the model to explicitly generate the steps of reasoning in natural language before arriving at the final answer. Other examples of prompting approaches include chain-of-thought with self-consistency

(Wang et al. (2023)), analogical reasoning (Yu et al. (2024)), and various modular approaches to address complex problems by decomposition to simpler sub-problems (Zhou et al. (2023); Khot et al. (2023); Creswell et al. (2023)). While these approaches show relative improvements in accuracy, the reasoning is still based on informal natural language and is prone to errors made by the LLMs in the steps of reasoning. In contrast, we follow the approach of off-loading the reasoning task to a formal solver that can guarantee correctness of the reasoning steps, and our particular focus is on the key challenge of ensuring that the correct formalization of the problem is sent to the solver.

**Tool-augmented reasoning.** Integrating LLMs with specialized tools for performing various tasks is becoming increasingly common (Schick et al. (2023)). This approach has also been adopted to improve the reasoning quality by augmenting the LLM with logical solvers or automated reasoning tools (Pan et al. (2023); Ye et al. (2023); Nye et al. (2021)). The key challenge with these approaches is to ensure that the LLM correctly translates the reasoning problem from NL to the formal language of the solver. This is the main focus of our work, where we show how verification and refinement with respect to concrete instantiations generated by the LLM can improve the translation accuracy and also provide a near-perfect precision of verification. Tool-augmented approaches have also been explored in the related areas of planning (Kambhampati et al. (2024); Guan et al. (2024)) and auto-formalization (Wu et al. (2022); Jiang et al. (2023); He-Yueya et al. (2023)), where informal mathematical proofs are translated to formal specifications defined in theorem provers like Isabelle (Paulson (1994)) and Lean (de Moura et al. (2015)). While our focus in this work has been on the general problem of logical reasoning, the core principle of verifying and refining formalizations with respect to concrete instantiations is also potentially applicable in these other domains.

**Self-verification approaches.** Many related works have also explored the notion of self-verification by LLMs (Weng et al. (2023); Madaan et al. (2023); Xie et al. (2023); Ling et al. (2023); Miao et al. (2024)). The general idea is that using the LLM to inspect and verify its own reasoning can show improvements, though in some domains self-critiquing has also shown diminished performance (Valmeekam et al. (2023)). Our approach of verification is different: instead of asking the LLM to verify the abstract chain of reasoning, we only ask it to generate concrete examples of the general constraints in the problem. The task of verification is then totally on the logical solver to formally check that these examples are consistent with the abstract formalization. Thus apart from not relying purely on the LLM for verification, we also avoid the more complex task of verifying an abstract chain of reasoning which can itself be highly error-prone. We instead perform both abstract and concrete inference and check consistency between them. We have shown how this approach can provide a very high precision verification, as opposed to the above approaches which provide relative improvements in accuracy. Our approach of inferring concrete instantiations is also similar to automated test case generation and verification in code generation approaches (Chen et al. (2024); Schäfer et al. (2024)). While our instantiations are similar to test cases, in general they can be arbitrary implications, and our focus is on logical expressions rather than code. Our approach also leverages compositionality as we infer instantiations for independent constraints identified from the problem, which can be seen as analogous to unit test generation in the code generation domain.

# 7 CONCLUSION

We have presented the Semantic Self-Verification approach, which substantially advances the robustness of AI reasoning systems by inferring verified problem formalizations through a novel combination of LLMs and logical solvers. Apart from boosting overall accuracy beyond the state-of-the-art, this approach introduces a novel verification feature that has near-perfect empirical precision.

As LLMs continue to evolve at a rapid pace, their reasoning abilities are becoming increasingly powerful. However, this general trend of improvement focuses on relative gains in answer accuracy on benchmarks, and when such benchmarks become saturated, more complex ones are introduced. While this ongoing progress is crucial, it does not inherently address the need for confidence of correctness on any arbitrary reasoning task. This is a key contribution of the SSV approach, which provides a complementary verification mechanism that is orthogonal to the underlying reasoning power of the particular LLM, and can hence be similarly applicable to more powerful models. As LLMs grow more capable, such a focus on *near-certain reasoning* through precise verification would be an important complimentary direction to general accuracy improvement—especially as we strive towards AI systems capable of super-human levels of reasoning.

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

# A    APPENDIX

## A.1    COMPOSITIONAL CODE GENERATION AND REFINEMENT PROMPTS

### A.1.1    PROBLEM DECOMPOSITION PROMPT

```
Given a problem description, please decompose it into an initial
context and a list of independent constraints.  If there is no
explicit initial context given and only constraints are given,
then just state "None" for initial context.  Some examples are
given below.
------
```
**Problem:**
```
The bald eagle eats the cow.  The bald eagle is red.  The bald
eagle needs the cow.  The bear needs the rabbit.  The cow is kind.
The cow is red.  The cow needs the bald eagle.  The rabbit eats
the bear.  The rabbit eats the cow.  The rabbit sees the cow.  If
something needs the bald eagle then it needs the rabbit.  If the
bald eagle is nice and the bald eagle is young then the bald eagle
sees the cow.  If the rabbit needs the cow then the cow sees the
rabbit.  If something eats the cow and the cow is nice then it
needs the bald eagle.  If something needs the rabbit then it is
nice.  If something sees the rabbit then it is red.  If something
needs the bald eagle then it eats the bald eagle.
```
**InitialContext:**
```
None
```
**Constraints:**
```
The bald eagle eats the cow.
```

```
###
The bald eagle is red.
###
The bald eagle needs the cow.
###
The bear needs the rabbit.
###
The cow is kind.
###
The cow is red.
###
The cow needs the bald eagle.
###
The rabbit eats the bear.
###
The rabbit eats the cow.
###
The rabbit sees the cow.
###
If something needs the bald eagle then it needs the rabbit.
###
If the bald eagle is nice and the bald eagle is young then the
bald eagle sees the cow.
###
If the rabbit needs the cow then the cow sees the rabbit.
###
If something eats the cow and the cow is nice then it needs the
bald eagle.
###
If something needs the rabbit then it is nice.
###
If something sees the rabbit then it is red.
###
If something needs the bald eagle then it eats the bald eagle.
------
```

**Problem:**
On Tuesday Vladimir and Wendy each eat exactly four separate
meals:  breakfast, lunch, dinner, and a snack.  The following
is all that is known about what they eat during that day:  At no
meal does Vladimir eat the same kind of food as Wendy.  Neither
of them eats the same kind of food more than once during the
day.  For breakfast, each eats exactly one of the following:  hot
cakes, poached eggs, or omelet.  For lunch, each eats exactly
one of the following:  fish, hot cakes, macaroni, or omelet.  For
dinner, each eats exactly one of the following:  fish, hot cakes,
macaroni, or omelet.  For a snack, each eats exactly one of the
following:  fish or omelet.  Wendy eats an omelet for lunch.

**InitialContext:**
On Tuesday Vladimir and Wendy each eat exactly four separate
meals:  breakfast, lunch, dinner, and a snack.

**Constraints:**
At no meal does Vladimir eat the same kind of food as Wendy.
###
Neither of them eats the same kind of food more than once during
the day.
###
For breakfast, each eats exactly one of the following:  hot cakes,
poached eggs, or omelet.
###

For lunch, each eats exactly one of the following:  fish, hot
cakes, macaroni, or omelet.
###
For dinner, each eats exactly one of the following:  fish, hot
cakes, macaroni, or omelet.
###
For a snack, each eats exactly one of the following:  fish or
omelet.
###
Wendy eats an omelet for lunch.
------

**Problem:**
In a repair facility there are exactly six technicians:  Stacy,
Urma, Wim, Xena, Yolanda, and Zane.  Each technician repairs
machines of at least one of the following three types-radios,
televisions, and VCRs-and no other types.  The following
conditions apply:  Xena and exactly three other technicians repair
radios.  Yolanda repairs both televisions and VCRs.  Stacy does
not repair any type of machine that Yolanda repairs.  Zane repairs
more types of machines than Yolanda repairs.  Wim does not repair
any type of machine that Stacy repairs.  Urma repairs exactly two
types of machines.

**InitialContext:**
In a repair facility there are exactly six technicians:  Stacy,
Urma, Wim, Xena, Yolanda, and Zane.  Each technician repairs
machines of at least one of the following three types-radios,
televisions, and VCRs-and no other types.

**Constraints:**
Xena and exactly three other technicians repair radios.
###
Yolanda repairs both televisions and VCRs.
###
Stacy does not repair any type of machine that Yolanda repairs.
###
Zane repairs more types of machines than Yolanda repairs.
###
Wim does not repair any type of machine that Stacy repairs.
###
Urma repairs exactly two types of machines.
------

A.1.2  INCREMENTAL CODE GENERATION PROMPT

Given a z3 program that models a particular problem and a new
constraint described in natural language, please provide the
z3 code to augment the program with the new constraint.  Please
provide only the z3 program code in the output and no other
markdown formatting or explanatory text.
------

**ExistingProgram:**
```
# On Tuesday Vladimir and Wendy each eat exactly four separate
meals:  breakfast, lunch, dinner, and a snack.
from z3 import *
people_sort, (Vladimir, Wendy) = EnumSort('people', ['Vladimir',
'Wendy'])
meals_sort, (breakfast, lunch, dinner, snack) = EnumSort('meals',
['breakfast', 'lunch', 'dinner', 'snack'])
foods_sort, (fish, hot_cakes, macaroni, omelet, poached_eggs) =
```

```
EnumSort('foods', ['fish', 'hot_cakes', 'macaroni', 'omelet',
'poached_eggs'])
people = [Vladimir, Wendy]
meals = [breakfast, lunch, dinner, snack]
foods = [fish, hot_cakes, macaroni, omelet, poached_eggs]
eats = Function('eats', people_sort, meals_sort, foods_sort)

pre_conditions = []

# CONSTRAINT: At no meal does Vladimir eat the same kind of food
as Wendy.
m = Const('m', meals_sort)
pre_conditions.append(ForAll([m], eats(Vladimir, m) != eats(Wendy,
m)))
```

**NewConstraint:**
Neither of them eats the same kind of food more than once during
the day.
**NewConstraintCode:**
```
m = Const('m', meals_sort)
p = Const('p', people_sort)
f = Const('f', foods_sort)
pre_conditions.append(ForAll([p, f], Sum([eats(p, m) == f for m in
meals]) <= 1))
------
```
**ExistingProgram:**
```
# In a repair facility there are exactly six technicians:  Stacy,
Urma, Wim, Xena, Yolanda, and Zane.  Each technician repairs
machines of at least one of the following three types|radios,
televisions, and VCRs|and no other types.
from z3 import *
technicians_sort, (Stacy, Urma, Wim, Xena, Yolanda, Zane)
= EnumSort('technicians', ['Stacy', 'Urma', 'Wim', 'Xena',
'Yolanda', 'Zane'])
machines_sort, (radios, televisions, VCRs) = EnumSort('machines',
['radios', 'televisions', 'VCRs'])
technicians = [Stacy, Urma, Wim, Xena, Yolanda, Zane]
machines = [radios, televisions, VCRs]
repairs = Function('repairs', technicians_sort, machines_sort,
BoolSort())

pre_conditions = []
t = Const('t', technicians_sort)
pre_conditions.append(ForAll([t], Sum([repairs(t, m) for m in
machines]) >= 1))
```

**NewConstraint:**
Xena and exactly three other technicians repair radios.
**NewConstraintCode:**
```
t = Const('t', technicians_sort)
pre_conditions.append(And(repairs(Xena, radios), Sum([And(t !=
Xena, repairs(t, radios)) for t in technicians]) == 3))
------
```

### A.1.3 OPTIONS CODE GENERATION PROMPT

```
Given a problem with multiple answer options and an existing
z3 program that models the problem, please provide the z3 code
that checks each option and prints the correct answer.  For
each option, first create the check_property for the option by
substituting the option values appropriately in the question
statement, as well as a full comment describing what the
check_property is stating.  Then use only the following custom
functions (is_unsat(), is_sat() and is_valid()) to check if the
check_property is unsatisfiable, satisfiable or valid (depending
on the question).  Please structure the code with comments
exactly as shown in the few shot examples below.  Please provide
only the options code and its comments in the output (not the
full program), and no other surrounding markdown formatting or
explanatory text.  Please create independently executable code
for each option (even if the option is not satisfiable) and do not
share code between different options.

def is_unsat(option_constraints):
solver = Solver()
solver.add(pre_conditions)
solver.add(option_constraints)
return solver.check() == unsat

def is_sat(option_constraints):
solver = Solver()
solver.add(pre_conditions)
return solver.check() == sat

def is_valid(option_constraints):
return is_sat(option_constraints) and is_unsat(Not(option_constraints))
------
```

>>> **Problem:**

```
On Tuesday Vladimir and Wendy each eat exactly four separate
meals:  breakfast, lunch, dinner, and a snack.
```

>>> **ExistingProgram:**

```
from z3 import *
people_sort, (Vladimir, Wendy) = EnumSort('people', ['Vladimir',
'Wendy'])
meals_sort, (breakfast, lunch, dinner, snack) = EnumSort('meals',
['breakfast', 'lunch', 'dinner', 'snack'])
foods_sort, (fish, hot_cakes, macaroni, omelet, poached_eggs) =
EnumSort('foods', ['fish', 'hot_cakes', 'macaroni', 'omelet',
'poached_eggs'])
people = [Vladimir, Wendy]
meals = [breakfast, lunch, dinner, snack]
foods = [fish, hot_cakes, macaroni, omelet, poached_eggs]
eats = Function('eats', people_sort, meals_sort, foods_sort)

pre_conditions = []

# CONSTRAINT: At no meal does Vladimir eat the same kind of food
as Wendy.
m = Const('m', meals_sort)
pre_conditions.append(ForAll([m], eats(Vladimir, m) != eats(Wendy,
```

```
1026  m)))
1027

1028
1029  # CONSTRAINT: Neither of them eats the same kind of food more than
      once during the day.
1030
1031  m = Const('m', meals_sort)
      p = Const('p', people_sort)
1032  f = Const('f', foods_sort)
1033  pre_conditions.append(ForAll([p, f], Sum([eats(p, m) == f for m in
1034  meals]) <= 1))

1035

1036
1037  >>> Question:
      Vladimir cannot eat which one of the following foods?
1038  >>> Options:
1039  (A) fish
1040  (B) hot cakes
1041  (C) macaroni
1042  (D) omelet
1043  (E) poached eggs
1044  >>> OptionsCode:

1045

1046  # CHECK TYPE: question says "cannot" so will check for validity
1047  using is_valid() to ensure that the negated statement is true in
1048  all possible models.

1049

1050  # OPTION A:
1051  # CHECK PROPERTY: Vladimir cannot eat which one of the following
1052  foods?  ANSWER: fish.
1053  m = Const('m', meals_sort)
1054  check_property = ForAll([m], eats(Vladimir, m) != fish)
1055  if is_valid(check_property):  print('(A)')

1056

1057  # OPTION B:
1058  # CHECK PROPERTY: Vladimir cannot eat which one of the following
1059  foods?  ANSWER: hot cakes.
1060  m = Const('m', meals_sort)
1061  check_property = ForAll([m], eats(Vladimir, m) != hot_cakes)
1062  if is_valid(check_property):  print('(B)')

1063

1064  # OPTION C:
1065  # CHECK PROPERTY: Vladimir cannot eat which one of the following
1066  foods?  ANSWER: macaroni.
1067  m = Const('m', meals_sort)
1068  check_property = ForAll([m], eats(Vladimir, m) != macaroni)
1069  if is_valid(check_property):  print('(C)')

1070

1071  # OPTION D:
1072  # CHECK PROPERTY: Vladimir cannot eat which one of the following
1073  foods?  ANSWER: omelet.
1074  m = Const('m', meals_sort)
1075  check_property = ForAll([m], eats(Vladimir, m) != omelet)
1076  if is_valid(check_property):  print('(D)')

1077

1078  # OPTION E:
1079  # CHECK PROPERTY: Vladimir cannot eat which one of the following
```

```
1080  foods?  ANSWER: poached eggs.
1081  m = Const('m', meals_sort)
1082  check_property = ForAll([m], eats(Vladimir, m) != poached_eggs)
1083  if is_valid(check_property):  print('(E)')
1084

1085  ______
1086  >>> Problem:
1087  In a repair facility there are exactly six technicians:  Stacy,
1088  Urma, Wim, Xena, Yolanda, and Zane.  Each technician repairs
1089  equipment of at least one of the following three types|radios,
1090  televisions, and VCRs|and no other types.
1091  >>> ExistingProgram:
1092  from z3 import *
1093  technicians_sort, (Stacy, Urma, Wim, Xena, Yolanda, Zane)
1094  = EnumSort('technicians', ['Stacy', 'Urma', 'Wim', 'Xena',
1095  'Yolanda', 'Zane'])
1096  equipment_sort, (radios, televisions, VCRs) =
1097  EnumSort('equipment', ['radios', 'televisions', 'VCRs'])
1098  technicians = [Stacy, Urma, Wim, Xena, Yolanda, Zane]
1099  equipment = [radios, televisions, VCRs]
1100  repairs = Function('repairs', technicians_sort, equipment_sort,
1101  BoolSort())

1102  pre_conditions = []
1103  t = Const('t', technicians_sort)
1104  pre_conditions.append(ForAll([t], Sum([repairs(t, e) for e in
1105  equipment]) >= 1))

1107  # CONSTRAINT: Xena and exactly three other technicians repair
1108  radios.
1109  t = Const('t', technicians_sort)
1110  pre_conditions.append(And(repairs(Xena, radios), Sum([And(t !=
1111  Xena, repairs(t, radios)) for t in technicians]) == 3))

1113  >>> Question:
1114  Which one of the following can be a complete and accurate list of
1115  the technicians that repair televisions?
1116  >>> Options:
1117  (A) Stacy, Wim, Zane
1118  (B) Urma, Wim, Xena, Yolanda
1119  (C) Xena, Yolanda
1120  (D) Stacy, Urma, Wim, Xena, Yolanda, Zane
1121  (E) Urma
1122  >>> OptionsCode:

1124  # CHECK TYPE: question says "can be" so will check for satisfiable
1125  using is_sat()

1127  # OPTION A:
1128  # CHECK PROPERTY: Which one of the following can be a complete and
1129  accurate list of the technicians that repair televisions?  ANSWER:
1130  Stacy, Wim and Zane.
1131  e = Const('e', equipment_sort)
1132  check_property = And(repairs(Stacy, televisions),
1133  repairs(Wim, televisions), repairs(Wim, televisions),
```

```
1134   Not(repairs(Urma, televisions)), Not(repairs(Xena, televisions)),
1135   Not(repairs(Yolanda, televisions)))
1136   if is_sat(check_property):  print('(A)')
1137
1138
1139   # OPTION B:
1140   # CHECK PROPERTY: Which one of the following can be a complete and
1141   accurate list of the technicians that repair televisions?  ANSWER:
       Urma, Wim, Xena and Yolanda.
1142   e = Const('e', equipment_sort)
1143   check_property = And(repairs(Urma, televisions), repairs(Wim,
1144   televisions), repairs(Xena, televisions), repairs(Yolanda,
1145   televisions), Not(repairs(Stacy, televisions)), Not(repairs(Zane,
1146   televisions)))
1147   if is_sat(check_property):  print('(B)')
1148
1149   # OPTION C:
1150   # CHECK PROPERTY: Which one of the following can be a complete and
1151   accurate list of the technicians that repair televisions?  ANSWER:
1152   Xena and Yolanda.
1153   e = Const('e', equipment_sort)
1154   check_property = And(repairs(Xena, televisions), repairs(Yolanda,
1155   televisions), Not(repairs(Stacy, televisions)), Not(repairs(Urma,
1156   televisions)), Not(repairs(Wim, televisions)), Not(repairs(Zane,
1157   televisions)))
1158   if is_sat(check_property):  print('(C)')
1159
1160   # OPTION D:
1161   # CHECK PROPERTY: Which one of the following can be a complete and
1162   accurate list of the technicians that repair televisions?  ANSWER:
1163   Stacy, Urma, Wim, Xena, Yolanda and Zane.
1164   e = Const('e', equipment_sort)
1165   check_property = And(repairs(Stacy, televisions), repairs(Urma,
1166   televisions), repairs(Wim, televisions), repairs(Xena,
1167   televisions), repairs(Yolanda, televisions), repairs(Zane,
       televisions))
1168   if is_sat(check_property):  print('(D)')
1169
1170   # OPTION E:
1171   # CHECK PROPERTY: Which one of the following can be a complete and
1172   accurate list of the technicians that repair televisions?  ANSWER:
1173   Urma.
1174   e = Const('e', equipment_sort)
1175   check_property = And(repairs(Urma, televisions),
1176   Not(repairs(Stacy, televisions)), Not(repairs(Wim, televisions)),
1177   Not(repairs(Xena, televisions)), Not(repairs(Yolanda,
1178   televisions)), Not(repairs(Zane, televisions)))
1179   if is_sat(check_property):  print('(E)')
1180   ------
```

### A.1.4 ERROR-BASED CODE REFINEMENT PROMPT

We are given a z3 program and an error message obtained from running it. First, please provide an analysis that investigates what may be the problem in the program that may be causing the error. Then, based on this analysis, please provide the corrected program where the issue is fixed – please make sure to retain any

```
comments from the original code in the repaired code (especially
the "CONSTRAINT", "QUESTION" or "OPTION" comments which demarcate
special code segments - please do not remove, change or add
any new such comments).  If there is a general issue in the
formulation, then please consider an alternative reformulation
so that the program can execute without errors.  A couple of
sample cases are shown below for illustration.  Please produce
output in exactly the format shown in these samples, with the ">>>
CorrectedProgram:" label clearly demarcating the corrected code,
and do not use any other markdown formatting.
------
```

>>> **ExistingProgram:**

```
from z3 import *
people_sort, (Vladimir, Wendy) = EnumSort('people', ['Vladimir',
'Wendy'])
meals_sort, (breakfast, lunch, dinner, snack) = EnumSort('meals',
['breakfast', 'lunch', 'dinner', 'snack'])
foods_sort, (fish, hot_cakes, macaroni, omelet, poached_eggs) =
EnumSort('foods', ['fish', 'hot_cakes', 'macaroni', 'omelet',
'poached_eggs'])
people = [Vladimir, Wendy]
meals = [breakfast, lunch, dinner, snack]
foods = [fish, hot_cakes, macaroni, omelet, poached_eggs]
eats = Function('eats', people_sort, meals_sort, foods_sort)

pre_conditions = []

# CONSTRAINT: At no meal does Vladimir eat the same kind of food
as Wendy.
pre_conditions.append(ForAll([m], eats(Vladimir, m) != eats(Wendy,
m)))
```

>>> **ErrorMessage:**

```
"NameError:  name 'm' is not defined"
```

>>> **ProblemDiscussion:**

This program defines three enumerations for people, meals, and
foods.  It then specifies that Vladimir and Wendy are people, and
lists the available meals and foods.  It also creates a function
eats which represents the food each person eats at each meal.
Finally, it tries to add a constraint to ensure that Vladimir and
Wendy do not eat the same kind of food at any meal.  However, as
the error message indicates, the constraint code uses a variable
'm' that has not been previously declared.  Hence the correct fix
to this issue would be to first explicitly declare the variable
'm' as a new const of meal_sort.

>>> **CorrectedProgram:**

```
from z3 import *
people_sort, (Vladimir, Wendy) = EnumSort('people', ['Vladimir',
'Wendy'])
meals_sort, (breakfast, lunch, dinner, snack) = EnumSort('meals',
['breakfast', 'lunch', 'dinner', 'snack'])
foods_sort, (fish, hot_cakes, macaroni, omelet, poached_eggs) =
EnumSort('foods', ['fish', 'hot_cakes', 'macaroni', 'omelet',
'poached_eggs'])
people = [Vladimir, Wendy]
meals = [breakfast, lunch, dinner, snack]
foods = [fish, hot_cakes, macaroni, omelet, poached_eggs]
eats = Function('eats', people_sort, meals_sort, foods_sort)
```

```
pre_conditions = []

# CONSTRAINT: At no meal does Vladimir eat the same kind of food
as Wendy.
m = Const('m', meals_sort)
pre_conditions.append(ForAll([m], eats(Vladimir, m) != eats(Wendy,
m)))
------
```

## A.2 INSTANTIATION GENERATION PROMPT

```
Given a problem scenario, some Z3 initialization code that
defines the data structures, and a list of constraints, please
provide positive and negative examples for each constraint.  Each
positive example should have a description and an expression of
concrete assignments that satisfy the constraint, while each
negative example should have a description and an expression
of concrete assignments that contradict the constraint.  If a
constraint or its examples cannot be expressed by the given
data structures or definitions, then please state "NONE" for the
example description and "pass" for the assignments code.  Please
provide the completion to the prompt in exactly the same format as
the example given below.
------
```
>>> **Scenario:**
```
None
```
>>> **InitializationCode:**
```
from z3 import *
creature_sort = DeclareSort('creature')
Stella = Const('Stella', creature_sort)
Jay = Const('Jay', creature_sort)
is_tumpus = Function('is_tumpus', creature_sort, BoolSort())
is_rompus = Function('is_rompus', creature_sort, BoolSort())
is_numpus = Function('is_numpus', creature_sort, BoolSort())
is_yumpus = Function('is_yumpus', creature_sort, BoolSort())
is_zumpus = Function('is_zumpus', creature_sort, BoolSort())
is_impus = Function('is_impus', creature_sort, BoolSort())
is_dumpus = Function('is_dumpus', creature_sort, BoolSort())
is_vumpus = Function('is_vumpus', creature_sort, BoolSort())
is_jompus = Function('is_jompus', creature_sort, BoolSort())
is_wumpus = Function('is_wumpus', creature_sort, BoolSort())
is_angry = Function('is_angry', creature_sort, BoolSort())
is_bright = Function('is_bright', creature_sort, BoolSort())
is_luminous = Function('is_luminous', creature_sort, BoolSort())
is_transparent = Function('is_transparent', creature_sort,
BoolSort())
is_bitter = Function('is_bitter', creature_sort, BoolSort())
is_red = Function('is_red', creature_sort, BoolSort())
is_happy = Function('is_happy', creature_sort, BoolSort())
is_large = Function('is_large', creature_sort, BoolSort())

pre_conditions = []
```
>>> **Constraints:**
```
Each dumpus is a vumpus.
###
Vumpuses are bright.
###
```

```
Every vumpus is a zumpus.
###
Zumpuses are not luminous.
>>> ConstraintExamples:
Constraint:
Each dumpus is a vumpus.
PositiveExampleDescription:
Stella is a dumpus and is also a vumpus.
PositiveExampleCode:
And(is_dumpus(Stella) == True, is_vumpus(Stella) == True)
NegativeExampleDescription:
Stella is a dumpus but is not a vumpus.
NegativeExampleCode:
And(is_dumpus(Stella) == True, is_vumpus(Stella) == False)
Constraint:
Vumpuses are bright.
PositiveExampleDescription:
Jay is a vumpus and is bright.
PositiveExampleCode:
And(is_vumpus(Jay) == True, is_bright(Jay) == True)
NegativeExampleDescription:
Jay is a vumpus and is not bright.
NegativeExampleCode:
And(is_vumpus(Jay) == True, is_bright(Jay) == False)
Constraint:
Every vumpus is a zumpus.
PositiveExampleDescription:
Jay is a vumpus and a zumpus.
PositiveExampleCode:
And(is_vumpus(Jay) == True, is_zumpus(Jay) == True)
NegativeExampleDescription:
Jay is a vumpus but not a zumpus.
NegativeExampleCode:
And(is_vumpus(Jay) == True, is_zumpus(Jay) == False)
Constraint:
Zumpuses are not luminous.
PositiveExampleDescription:
Stella is a zumpus and is not luminous.
PositiveExampleCode:
And(is_zumpus(Stella) == True, is_luminous(Stella) == False)
NegativeExampleDescription:
Stella is a zumpus and is luminous.
NegativeExampleCode:
And(is_zumpus(Stella) == True, is_luminous(Stella) == True)
------
>>> Scenario:
On Tuesday Vladimir and Wendy each eat exactly two separate meals:
breakfast and dinner.
>>> InitializationCode:
from z3 import *

people_sort, (Vladimir, Wendy) = EnumSort('people', ['Vladimir',
'Wendy'])
meals_sort, (breakfast, dinner) = EnumSort('meals', ['breakfast',
'dinner'])
foods_sort, (fish, hot_cakes, macaroni, omelet, poached_eggs) =
EnumSort('foods', ['fish', 'hot_cakes', 'macaroni', 'omelet',
'poached_eggs'])
```

```
people = [Vladimir, Wendy]
meals = [breakfast, dinner]
foods = [fish, hot_cakes, macaroni, omelet, poached_eggs]
eats = Function('eats', people_sort, meals_sort, foods_sort)

pre_conditions = []
```
>>> **Constraints:**
```
At no meal does Vladimir eat the same kind of food as Wendy.
###
Neither of them eats the same kind of food more than once during
the day.
###
For breakfast, each eats hot cakes.
```
>>> **ConstraintExamples:**
```
Constraint:
At no meal does Vladimir eat the same kind of food as Wendy.
PositiveExampleDescription:
Vladimir and Wendy eat different foods at each meal:  Vladimir
has fish for breakfast while Wendy has hot cakes, and for dinner,
Vladimir eats macaroni while Wendy has omelet.
PositiveExampleCode:
And(eats(Vladimir, breakfast) == fish, eats(Wendy, breakfast) ==
hot_cakes,
eats(Vladimir, dinner) == macaroni, eats(Wendy, dinner) == omelet)
NegativeExampleDescription:
At dinner, both Vladimir and Wendy eat the same food, macaroni.
NegativeExampleCode:
And(eats(Vladimir, dinner) == macaroni, eats(Wendy, dinner) ==
macaroni)
Constraint:
Neither of them eats the same kind of food more than once during
the day.
PositiveExampleDescription:
Vladimir eats different foods for breakfast and dinner:  fish for
breakfast and hot cakes for dinner.  Wendy also eats different
foods for both meals:  hot cakes for breakfast and omelet for
dinner.
PositiveExampleCode:
And(eats(Vladimir, breakfast) == fish, eats(Vladimir, dinner) ==
hot_cakes,
eats(Wendy, breakfast) == hot_cakes, eats(Wendy, dinner) ==
omelet)
NegativeExampleDescription:
Vladimir eats fish for both breakfast and dinner.
NegativeExampleCode:
And(eats(Vladimir, breakfast) == fish, eats(Vladimir, dinner) ==
fish)
Constraint:
For breakfast, each eats hot cakes.
PositiveExampleDescription:
Vladimir and Wendy both eat hot cakes for breakfast.
PositiveExampleCode:
And(eats(Vladimir, breakfast) == hot_cakes, eats(Wendy, breakfast)
== hot_cakes)
NegativeExampleDescription:
Vladimir eats macaroni for breakfast.
NegativeExampleCode:
eats(Vladimir, breakfast) == macaroni
```

```
------
>>> Scenario:
In a repair facility there are exactly six technicians: Stacy,
Urma, Wim, Xena, Yolanda, and Zane. Each technician repairs
machines of at least one of the following three types|radios,
televisions, and VCRs|and no other types.
>>> InitializationCode:
from z3 import *
technicians_sort, (Stacy, Urma, Wim, Xena, Yolanda, Zane)
= EnumSort('technicians', ['Stacy', 'Urma', 'Wim', 'Xena',
'Yolanda', 'Zane'])
machines_sort, (radios, televisions, VCRs) = EnumSort('machines',
['radios', 'televisions', 'VCRs'])
technicians = [Stacy, Urma, Wim, Xena, Yolanda, Zane]
machines = [radios, televisions, VCRs]
repairs = Function('repairs', technicians_sort, machines_sort,
BoolSort())

pre_conditions = []
t = Const('t', technicians_sort)
pre_conditions.append(ForAll([t], Sum([repairs(t, m) for m in
machines]) >= 1))

>>> Constraints:
Xena and exactly three other technicians repair radios.
###
Stacy needs help repairing VCRs.
###
Urma and Zane repair the same type of machine.
>>> ConstraintExamples:
Constraint:
Xena and exactly three other technicians repair radios.
PositiveExampleDescription:
Only Xena, Wim, Yolanda, and Zane repair radios and no one else.
PositiveExampleCode:
And(repairs(Stacy, radios) == False, repairs(Urma, radios) ==
False, repairs(Wim, radios) == True, repairs(Xena, radios) ==
True, repairs(Yolanda, radios) == True, repairs(Zane, radios) ==
True)
NegativeExampleDescription:
Only Xena and Yolanda repair radios and no one else.
NegativeExampleCode:
And(repairs(Stacy, radios) == False, repairs(Urma, radios) ==
False, repairs(Wim, radios) == False, repairs(Xena, radios) ==
True, repairs(Yolanda, radios) == True, repairs(Zane, radios) ==
False)
Constraint:
Stacy needs help repairing VCRs.
PositiveExampleDescription:
NONE
PositiveExampleCode:
pass
NegativeExampleDescription:
NONE
NegativeExampleCode:
pass
Constraint:
Urma and Zane repair the same type of machine.
```

```
PositiveExampleDescription:
Urma and Zane both repair VCRs.
PositiveExampleCode:
And(repairs(Urma, VCRs) == True, repairs(Zane, VCRs) == True)
NegativeExampleDescription:
Urma repairs televisions, while Zane repairs radios.
NegativeExampleCode:
And(repairs(Urma, televisions) == True, repairs(Zane, radios) ==
True)
------
```

## A.3 SEMANTIC REPAIR PROMPT

```
We are given a scenario description, some initial z3 code that
sets up basic definitions, a constraint in natural language,
and a code snippet that implements that constraint.  We are also
given some code that should implement a positive example to the
constraint, which should be satisfiable under that constraint, but
it is not.  First, please provide an analysis that investigates
what may be the problem in either the initial code, the constraint
code or the example.  Then, based on this analysis, please repair
the relevant code segments (initial code, constraint code, or
example code) so that the positive example becomes satisfiable
(state 'NONE' if no repair is required to a code segment).  If
multiple segments are incorrect due to a general formulation
problem, then please reformulate the whole solution approach in
the initial code and produce appropriate code for all segments.  A
couple of sample cases are shown below for illustration.  Please
produce output in exactly the format shown in these samples, and
do not use any other markdown formatting.
------
```
**Scenario:**
```
On Tuesday Vladimir and Wendy each eat exactly four separate
meals:  breakfast, lunch, dinner, and a snack.
```
**InitialCode:**
```
from z3 import *
people_sort, (Vladimir, Wendy) = EnumSort('people', ['Vladimir',
'Wendy'])
meals_sort, (breakfast, lunch, dinner, snack) = EnumSort('meals',
['breakfast', 'lunch', 'dinner', 'snack'])
foods_sort, (fish, hot_cakes, macaroni, omelet, poached_eggs) =
EnumSort('foods', ['fish', 'hot_cakes', 'macaroni', 'omelet',
'poached_eggs'])
people = [Vladimir, Wendy]
meals = [breakfast, lunch, dinner, snack]
foods = [fish, hot_cakes, macaroni, omelet, poached_eggs]
eats = Function('eats', people_sort, meals_sort, foods_sort)

pre_conditions = []
```

**ConstraintDescription:**
```
At some meal Vladimir eats the same kind of food as Wendy.
```
**ConstraintCode:**
```
m = Const('m', meals_sort)
pre_conditions.append(ForAll([m], eats(Vladimir, m) != eats(Wendy,
m)))
```
**PositiveExampleCode:**

```
And(eats(Vladimir, breakfast) == fish, eats(Wendy, breakfast) ==
fish)
```

**ProblemDiscussion:**
The scenario describes foods that Vladimir and Wendy eat at
various meals during the day. The initial code defines the
main data structures and the eats function which indicates the
food each person eats on every meal. The constraint requires
that there is at least one meal where they both eat the same
food. The constraint code asserts that for all meals, the food
that Vladimir eats is different from what Wendy eats. But this
contradicts the intended constraint. The positive example code
states that at breakfast, both Vladimir and Wendy eat fish, and
this is consistent with the requirements of the constraint. Hence
there is no issue in the initial code and the example code, but
the constraint code wrongly implements the constraint. It should
be repaired to assert that for some meal, both Vladimir and Wendy
eat the same food.

**RepairedInitialCode:**
```
NONE
```
**RepairedConstraintCode:**
```
m = Const('m', meals_sort)
pre_conditions.append(Exists([m], eats(Vladimir, m) == eats(Wendy,
m)))
```
**RepairedPositiveExampleCode:**
```
NONE
------
```
**Scenario:**
In a repair facility there are exactly six technicians: Stacy,
Urma, Wim, Xena, Yolanda, and Zane. Each technician repairs
machines of at least one of the following three types|radios,
televisions, and VCRs|and no other types.

**InitialCode:**
```
from z3 import *
technicians_sort, (Stacy, Urma, Wim, Xena, Yolanda, Zane)
= EnumSort('technicians', ['Stacy', 'Urma', 'Wim', 'Xena',
'Yolanda', 'Zane'])
machines_sort, (radios, televisions, VCRs) = EnumSort('machines',
['radios', 'televisions', 'VCRs'])
technicians = [Stacy, Urma, Wim, Xena, Yolanda, Zane]
machines = [radios, televisions, VCRs]
repairs = Function('repairs', technicians_sort, machines_sort,
BoolSort())

pre_conditions = []
t = Const('t', technicians_sort)
pre_conditions.append(ForAll([t], Sum([repairs(t, m) for m in
machines]) <= 1))
```
**ConstraintDescription:**
Urma repairs radios and VCRs

**ConstraintCode:**
```
pre_conditions.append(And(repairs(Urma, radios), repairs(Urma,
VCRs)))
```
**PositiveExampleCode:**
```
And(repairs(Urma, radios) == True, repairs(Urma, VCRs) == True)
```
**ProblemDiscussion:**
The scenario describes types of machines that technicians repair
at a repair facility, where each technician repairs at least
one type of machine. The initial code defines the main data

structures and the repairs function which indicates the type of machine repaired by each technician. It also adds the general condition that each technician can repair at most one type of machine, which is an incorret interpretation of the scenario statement that each technician must repair AT LEAST one type of machine. The constraint requires that Urma repairs both VCRs and radios. The constraint code correctly asserts this requirement, and the positive example code also states this correctly. Hence there is no issue in the constraint code and the example code, but the initial code wrongly prevents any technician from repairing two kinds of machines. It should be repaired to assert that each technician must repair at least one kind of machine.

**RepairedInitialCode:**

```
from z3 import *
technicians_sort, (Stacy, Urma, Wim, Xena, Yolanda, Zane)
= EnumSort('technicians', ['Stacy', 'Urma', 'Wim', 'Xena',
'Yolanda', 'Zane'])
machines_sort, (radios, televisions, VCRs) = EnumSort('machines',
['radios', 'televisions', 'VCRs'])
technicians = [Stacy, Urma, Wim, Xena, Yolanda, Zane]
machines = [radios, televisions, VCRs]
repairs = Function('repairs', technicians_sort, machines_sort,
BoolSort())

pre_conditions = []
t = Const('t', technicians_sort)
pre_conditions.append(ForAll([t], Sum([repairs(t, m) for m in
machines]) >= 1))
```

**RepairedConstraintCode:**

```
NONE
```

**RepairedPositiveExampleCode:**

```
NONE
------
```

### A.4 DATASET CORRECTION CASES

We found a small number of cases in three of the datasets where the answers have been labelled incorrectly. Our SSV system (with GPT-4 base model) detected these cases in its verification, and we describe the corrections that should be made to the datasets below.

#### A.4.1 AR-LSAT CORRECTIONS

Three cases in the AR-LSAT dataset were verified correctly by our system, but were labelled with the wrong answers in the dataset. These three cases are **ar_lsat_201612_3-G_2_6** (correct answer should be D but incorrectly labelled C), **ar_lsat_201612_3-G_1_4** (correct answer should be E but incorrectly labelled A) and **ar_lsat_201612_3-G_2_8** (correct answer should be B but is incorrectly labelled A). For all three of these cases, we were able to check the reasoning and also that the answers in the original source LSAT Test (https://img.cracklsat.net/lsat/pt/pt80.pdf) are consistent with the answers that were generated by our system. Hence we submit that these are errors in the AR-LSAT dataset collection process.

#### A.4.2 FOLIO CORRECTIONS

In the FOLIO dataset, we found one case that was correctly verified by our system, but we find is labelled with the wrong answer in the dataset. This is case **FOLIO_dev_27**:

*All aliens are extraterrestrial. If someone is from Mars, then they are aliens. No extraterrestrial is human. Everyone from Earth is a human. Marvin cannot be from Earth and from Mars. If Marvin is*

*not from Earth, then Marvin is an extraterrestrial. Based on the above information, is the following statement true, false, or uncertain? Marvin is an alien.*

We submit that the correct answer is C (unknown) but it is labelled B (false) in the dataset. Reasoning: If Marvin is from Earth, he is not an alien. If Marvin is not from Earth: If he is from Mars, he is an alien, otherwise, we cannot be certain he is an alien. Hence both outcomes are possible.

We suspect the error in the dataset may stem from an incorrect formalization of the problem in the original FOLIO dataset source:https://github.com/Yale-LILY/FOLIO/blob/main/data/v0.0/folio-validation.txt. In this source we see that the constraint "Marvin cannot be from Earth and from Mars" is incorrectly formalized as $\neg FromEarth(marvin) \wedge \neg FromMars(marvin)$ in first order logic, which asserts that Marvin is neither from Earth nor from Mars.

### A.4.3 PROOFWRITER CORRECTIONS

In the ProofWriter dataset, we found 6 cases that were correctly verified by our system, but we find are labelled with the wrong answer in the dataset. In all 6 cases, the answers in the dataset have been labelled as unknown when they can be proven to be either true or false as we show below.

**ProofWriter_RelNeg-OWA-D5-450_Q22** (Correct answer should be B (false), but labelled C (unknown)).

*The bald eagle chases the lion. The bald eagle is not green. The bald eagle is round. The bald eagle likes the lion. The dog is red. The lion does not chase the dog. The lion is round. The lion is not young. The rabbit chases the dog. The rabbit eats the lion. If something chases the dog then it likes the rabbit. If something is red and it chases the lion then the lion likes the bald eagle. If something is big then it chases the rabbit. If something is round and it chases the bald eagle then the bald eagle does not like the dog. If something likes the lion then it is red. If something is red and round then it does not chase the bald eagle. If something is red and young then it chases the bald eagle. If something likes the bald eagle and the bald eagle chases the lion then it likes the lion. If something eats the bald eagle then the bald eagle is red. Based on the above information, is the following statement true, false, or unknown? The bald eagle is young.*

Reasoning:

From Fact 4 and Rule 5:

The bald eagle likes the lion. Therefore, the bald eagle is red.

From Fact 3:

The bald eagle is round. Applying Rule 6 to the bald eagle:

The bald eagle is red and round. Therefore, the bald eagle does not chase itself. Assuming the bald eagle is young:

The bald eagle is red and young. Applying Rule 7 to the bald eagle:

The bald eagle is red and young. Therefore, the bald eagle chases itself. Contradiction:

From step 3, the bald eagle does not chase itself.

From step 5, the bald eagle chases itself.

This is a contradiction.

Conclusion: The assumption that the bald eagle is young leads to a contradiction. Therefore, the bald eagle cannot be young.

**ProofWriter_AttNeg-OWA-D5-471_Q14** (Correct answer should be A (true), but labelled C (unknown)).

*Anne is white. Charlie is cold. Charlie is round. Charlie is young. Gary is kind. Gary is nice. Gary is round. Gary is white. Gary is young. Harry is blue. Harry is cold. Harry is kind. Harry is white. Harry is young. White, kind things are blue. If something is white then it is kind. Nice things are kind. All blue, nice things are young. All blue, white things are nice. If something is round and*

*not nice then it is not cold. Blue, young things are cold. Based on the above information, is the following statement true, false, or unknown? Charlie is kind.*

Reasoning:

Relevant facts: Charlie is cold. Charlie is round. Charlie is young.

Relevant Rules:

If something is round and not nice, then it is not cold. (Rule 6)

Nice things are kind. (Rule 3)

Assuming Charlie is not nice:

Since Charlie is round and not nice, according to Rule 6, Charlie should not be cold. However, this contradicts the fact that Charlie is cold. Therefore, our assumption that Charlie is not nice must be false.

Conclusion from the contradiction: Charlie must be nice.

Applying Rule 3:

Since nice things are kind, and Charlie is nice, it follows that Charlie is kind.

**ProofWriter_AttNeg-OWA-D5-112_Q20**   (Correct answer should be B (false), but labelled C (unknown)).

*Charlie is kind. Charlie is nice. Charlie is quiet. Dave is rough. Dave is white. Erin is nice. Gary is not white. If something is cold then it is not furry. If Charlie is quiet then Charlie is nice. Kind things are white. Nice things are kind. If something is rough then it is kind. Cold, quiet things are rough. All cold things are quiet. If something is white and nice then it is cold. If Erin is cold then Erin is nice. Based on the above information, is the following statement true, false, or unknown? Gary is nice.*

Reasoning:

Gary is not white. (rule 1)

Nice things are kind. (rule 2)

Kind things are white. (rule 3)

If Gary were nice, then by rule 2, he would also be kind. If Gary is kind, then by rule 3, he must be white. However, rule 1 tells us that Gary is not white. This creates a contradiction because Gary cannot be both not white and white at the same time.

Given that Gary is not white, he cannot be kind, and therefore, he cannot be nice. Thus, the statement "Gary is nice" is false.

**ProofWriter_AttNeg-OWA-D5-850_Q14**   (Correct answer should be B (false), but labelled C (unknown)).

*Anne is red. Anne is smart. Bob is kind. Bob is not nice. Fiona is furry. Fiona is rough. Gary is not green. Gary is kind. Gary is nice. Gary is rough. If someone is nice then they are red. Smart people are green. If someone is smart and red then they are not kind. All rough, green people are nice. Green people are rough. If someone is red and green then they are rough. If someone is furry and green then they are smart. All rough, furry people are smart. Furry, rough people are smart. Based on the above information, is the following statement true, false, or unknown? Bob is smart.*

Reasoning:

Bob is kind. Bob is not nice.

Rule: Smart people are green. So, if Bob were smart, he would be green.

Rule: Green people are rough. Therefore, if Bob were green (and thus rough), we can use the next rule.

Rule: All rough, green people are nice. If Bob were rough and green, he would be nice, but we know Bob is not nice.

Conclusion: Bob cannot be green because it would contradict the fact that he is not nice. Since Bob is not green, and smart people are green, Bob cannot be smart.

**ProofWriter_AttNeg-OWA-D5-219_Q13**   (Correct answer should be A (true), but labelled C (unknown)).

*Charlie is not quiet. Dave is big. Dave is furry. Erin is cold. Erin is not green. Erin is not kind. Fiona is quiet. Big things are young. Young, cold things are big. Quiet things are big. All young things are cold. If something is big and not furry then it is cold. If something is cold then it is not kind. If something is cold and big then it is quiet. If Fiona is cold and Fiona is not quiet then Fiona is kind. If something is quiet and not kind then it is green. Based on the above information, is the following statement true, false, or unknown? Charlie is not big.*

Reasoning:

Charlie is not quiet.

Assume for contradiction that Charlie is big.

Big things are young: Therefore, Charlie is young.

All young things are cold: Therefore, Charlie is cold.

If something is cold, then it is not kind: Therefore, Charlie is not kind.

If something is cold and big, then it is quiet: Therefore, Charlie is quiet.

This contradicts the given fact that Charlie is not quiet. Therefore, Charlie is not big.

**ProofWriter_AttNeg-OWA-D5-94_Q18**   (Correct answer should be B (false), but labelled C (unknown))

*Bob is smart. Charlie is kind. Charlie is not smart. Fiona is blue. Fiona is rough. Fiona is smart. Gary is kind. All cold, quiet people are smart. If someone is cold then they are smart. If someone is red and kind then they are smart. If someone is quiet then they are blue. If someone is blue then they are quiet. If someone is kind then they are rough. If Gary is kind and Gary is rough then Gary is quiet. All blue, smart people are red. Blue, rough people are red. Based on the above information, is the following statement true, false, or unknown? Charlie is blue.*

Reasoning:

Charlie is kind.

If someone is kind, then they are rough

Therefore, Charlie is rough.

Assume for contradiction that Charlie is blue.

Blue, rough people are red. Since Charlie is both blue (assumed) and rough, Charlie must be red.

If someone is red and kind, then they are smart.

Since Charlie is red (from step 4) and kind (from step 1), Charlie must be smart.

However, it's given that Charlie is not smart (from the context).

Hence, we have a contradiction. Therefore, Charlie is not blue.

A.5   ANALYSIS OF VERIFICATION FAILURE CASES

We conducted a manual analysis over a sample of 30 cases where SSV verification failed. Here is a summary of the failure reasons:

- code incorrect, example correct: 16 (53.3%)

- code incorrect, example incorrect: 7 (23.3%)

- code correct, example incorrect: 3 (10%)

- program not well-formed: 4 (13.3%)

We see that in most cases the code is incorrect as opposed to examples, which can be expected as examples inference is generally simpler than abstract translation. Below is the detailed analysis of the reasons for the verification failure for specific cases.

**InitialContext:**

```
A bakery makes exactly three kinds of cookie|oatmeal, peanut
butter, and sugar.
Exactly three batches of each kind of cookie are made each week
(Monday through Friday)
and each batch is made, from start to finish, on a single day.
```

**InitialCode:**

```
from z3 import *

days_sort, (Monday, Tuesday, Wednesday, Thursday, Friday) =
EnumSort('days', ['Monday', 'Tuesday', 'Wednesday', 'Thursday',
'Friday'])
cookies_sort, (oatmeal, peanut_butter, sugar) =
EnumSort('cookies', ['oatmeal', 'peanut_butter', 'sugar'])
batch_number_sort = IntSort()
make_batch = Function('make_batch', cookies_sort,
batch_number_sort, days_sort)

pre_conditions = []
batches = range(1, 3+1)  # Corrected to only three batches
days = [Monday, Tuesday, Wednesday, Thursday, Friday]
cookies = [oatmeal, peanut_butter, sugar]

for cookie in cookies:
    for b in batches:
        d = Const('d', days_sort)  # Declare a constant of type
        days_sort
        pre_conditions.append(Exists([d], make_batch(cookie, b) ==
        d))
```

**NegExample:**

```
And(make_batch(oatmeal, 1) == Monday, make_batch(oatmeal, 2) ==
Tuesday, make_batch(oatmeal, 3) == Wednesday, make_batch(oatmeal,
4) == Thursday)
```

**NegExampleDescription:**

```
Four batches of oatmeal cookies are made, which contradicts the
constraint that exactly three batches of each kind of cookie are
made each week.
```

**Analysis:**

```
Code incorrect, Example correct. It does not implement the
intended constraint that only 3 batches of each kind of cookie
should be made in a week, which the negative example correctly
tests.
```

**InitialContext:**

A loading dock consists of exactly six bays numbered 1 through 6 consecutively from one side of the dock to the other. Each bay is holding a different one of exactly six types of cargo|fuel, grain, livestock, machinery, produce, or textiles.

**InitialCode:**

```
from z3 import *

bays_sort = IntSort()
cargo_sort, (fuel, grain, livestock, machinery, produce, textiles)
= EnumSort('cargo', ['fuel', 'grain', 'livestock', 'machinery',
'produce', 'textiles'])
cargo = [fuel, grain, livestock, machinery, produce, textiles]
holding = Function('holding', bays_sort, cargo_sort)

pre_conditions = []

pre_conditions.append(Distinct([holding(b) for b in range(1, 7)]))
pre_conditions.append(And([holding(b) != None for b in range(1,
7)]))
```

**Condition:**

The bay holding textiles is next to the bay holding produce.

**ConditionCode:**

```
b = Int('b')
pre_conditions.append(Or(holding(b) == textiles, holding(b+1) ==
produce, holding(b-1) == produce))
```

**NegExample:**

```
And(holding(1) == textiles, holding(4) == produce)
```

**NegExampleDescription:**

Textiles are in bay 1 and produce is in bay 4.

**Analysis:**

Code incorrect, Example correct. It does not correctly implement the constraint that textiles should be right next to produce as it is only a disjunction of the three conditions. The negative example correctly checks this condition by allowing textiles in bay 1 and produce in bay 4.

---

**InitialContext:**

On each of exactly seven consecutive days (day 1 though day 7), a pet shop features exactly one of three breeds of kitten|Himalayan, Manx, Siamese|and exactly one of three breeds of puppy|Greyhound, Newfoundland, Rottweiler.

**InitialCode:**

```
from z3 import *

days_sort, (day1, day2, day3, day4, day5, day6, day7) =
EnumSort('days', ['day1', 'day2', 'day3', 'day4', 'day5', 'day6',
'day7'])
kitten_breeds_sort, (Himalayan, Manx, Siamese) =
EnumSort('kitten_breeds', ['Himalayan', 'Manx', 'Siamese'])
puppy_breeds_sort, (Greyhound, Newfoundland, Rottweiler) =
EnumSort('puppy_breeds', ['Greyhound', 'Newfoundland',
'Rottweiler'])
days = [day1, day2, day3, day4, day5, day6, day7]
kitten_breeds = [Himalayan, Manx, Siamese]
puppy_breeds = [Greyhound, Newfoundland, Rottweiler]

features_kitten = Function('features_kitten', days_sort,
kitten_breeds_sort, BoolSort())
features_puppy = Function('features_puppy', days_sort,
puppy_breeds_sort, BoolSort())

pre_conditions = []

# Add constraints to ensure exactly one breed of kitten and one
breed of puppy is featured each day
for day in days:
    pre_conditions.append(Sum([If(features_kitten(day, breed), 1,
    0) for breed in kitten_breeds]) == 1)
    pre_conditions.append(Sum([If(features_puppy(day, breed), 1,
    0) for breed in puppy_breeds]) == 1)
```

**Condition:**

No breed is featured on any two consecutive days.

**ConditionCode:**

```
d1 = Const('d1', days_sort)
d2 = Const('d2', days_sort)
kb = Const('kb', kitten_breeds_sort)
pb = Const('pb', puppy_breeds_sort)
pre_conditions.append(ForAll([d1, d2, kb],
Implies(And(features_kitten(d1, kb), features_kitten(d2, kb)), d1
== d2)))
pre_conditions.append(ForAll([d1, d2, pb],
Implies(And(features_puppy(d1, pb), features_puppy(d2, pb)), d1 ==
d2)))
```

**PosExample:**

```
And(features_kitten(day2, Siamese) == True, features_kitten(day3,
Manx) == True)
```

**PosExampleDescription:**

```
Siamese kittens are featured on day 2 and Manx kittens are
featured on day 3.
```

**Analysis:**

```
Code incorrect, Example correct. The condition code does not
implement consecutiveness constraint but instead requires each
breed to only be featured on one day, which is not a requirement
and causes the positive example to fail. The positive example
correctly presents a valid instantiation of the problem.
```

---

**InitialContext:**

```
The organizer of a reading club will select at least five and at
most six works from a group of nine works. The group consists of
three French novels, three Russian novels, two French plays, and
one Russian play.
```

**InitialCode:**

```
from z3 import *

works_sort, (french_novel1, french_novel2, french_novel3,
russian_novel1, russian_novel2, russian_novel3, french_play1,
french_play2, russian_play) = EnumSort('works', ['french_novel1',
'french_novel2', 'french_novel3', 'russian_novel1',
'russian_novel2', 'russian_novel3', 'french_play1',
'french_play2', 'russian_play'])
works = [french_novel1, french_novel2, french_novel3,
russian_novel1, russian_novel2, russian_novel3, french_play1,
french_play2, russian_play]
selected = Function('selected', works_sort, BoolSort())

pre_conditions = []
```

**NegExample:**

```
And(selected(french_novel1) == True, selected(french_novel2) ==
True, selected(french_novel3) == True, selected(russian_novel1) ==
True, selected(russian_novel2) == True, selected(russian_novel3)
== True, selected(french_play1) == True, selected(french_play2) ==
True, selected(russian_play) == True)
```

**NegExampleDescription:**

```
The organizer selects all three French novels, all three Russian
novels, both French plays, and the Russian play, which exceeds the
maximum of six works.
```

**Analysis:**

```
Code incorrect, Example correct. The negative example correctly
violates the constraint of not exceeding 6 works, but the code
does not implement this constraint.
```

---

**InitialContext:**

```
A bakery makes exactly three kinds of cookie|oatmeal, peanut
butter, and sugar.
Exactly three batches of each kind of cookie are made each week
(Monday through Friday)
and each batch is made, from start to finish, on a single day.
```

**InitialCode:**

```
1998   from z3 import *
1999
2000   days_sort, (Monday, Tuesday, Wednesday, Thursday, Friday) =
2001   EnumSort('days', ['Monday', 'Tuesday', 'Wednesday', 'Thursday',
2002   'Friday'])
2003   cookies_sort, (oatmeal, peanut_butter, sugar) =
2004   EnumSort('cookies', ['oatmeal', 'peanut_butter', 'sugar'])
2005   batch_sort = IntSort()
2006   make_batch = Function('make_batch', cookies_sort, batch_sort,
2007   days_sort, BoolSort())
2008   pre_conditions = []
2009
2010   # Add constraints to ensure exactly three batches of each kind of
2011   cookie are made each week
2011   for cookie in [oatmeal, peanut_butter, sugar]:
2012       pre_conditions.append(Sum([If(make_batch(cookie, i, d), 1, 0)
2013       for i in range(1, 4) for d in [Monday, Tuesday, Wednesday,
2014       Thursday, Friday]]) == 3)
```

**NegExample:**

```
And(make_batch(oatmeal, 1, Monday) == True, make_batch(oatmeal, 2,
Tuesday) == True, make_batch(oatmeal, 3, Wednesday) == True,
make_batch(oatmeal, 4, Thursday) == True)
```

**NegExampleDescription:**

Four batches of oatmeal cookies are made, which contradicts the
constraint that exactly three batches of each kind of cookie are
made each week.

**Analysis:**

Code incorrect, Example correct. The negative example correctly
violates the constraint by enforcing 4 batches oatmeal cookies to
be made in the week.

---

**InitialContext:**

An administrator must assign parking spaces to six new employees:
Robertson, Souza, Togowa, Vaughn, Xu, and Young.
Each of the six employees must be assigned one of the following
parking spaces: #1, #2, #3, #4, #5, or #6.
No two employees can be assigned the same parking space.

**InitialCode:**

```
from z3 import *

employees_sort, (Robertson, Souza, Togowa, Vaughn, Xu, Young) =
EnumSort('employees', ['Robertson', 'Souza', 'Togowa', 'Vaughn',
'Xu', 'Young'])
parking_spaces_sort = IntSort()
parking_space = Function('parking_space', employees_sort,
parking_spaces_sort)
pre_conditions = []

pre_conditions.append(Distinct([parking_space(e) for e in
[Robertson, Souza, Togowa, Vaughn, Xu, Young]]))
```

```
pre_conditions.append(And([parking_space(e) >= 1 for e in
[Robertson, Souza, Togowa, Vaughn, Xu, Young]]))
pre_conditions.append(And([parking_space(e) <= 6 for e in
[Robertson, Souza, Togowa, Vaughn, Xu, Young]]))
```

**Condition:**

If Togowa is assigned a higher-numbered parking space than Souza,
then Togowa is assigned parking space #3.

**ConditionCode:**

```
additional_constraint = parking_space(Togowa) >
parking_space(Souza)
check_property = And(additional_constraint, parking_space(Togowa)
== 3)
pre_conditions.append(check_property)
```

**ExampleFail:**

Positive example not inferred (NONE description)

**Analysis:**

Code incorrect, Example incorrect. The code does not implement the
conditional requirement but only a conjunction. A positive example
could not be inferred from the LLM or valid solver code
implementing the positive example.

---

**InitialContext:**

A panel of five scientists will be formed. The panelists will be
selected from among three botanists|F, G, and H|three chemists|K,
L, and M|and three zoologists|P, Q, and R.

**InitialCode:**

```
from z3 import *

scientist_sort = DeclareSort('scientist')
F, G, H, K, L, M, P, Q, R = Consts('F G H K L M P Q R',
scientist_sort)
botanists = [F, G, H]
chemists = [K, L, M]
zoologists = [P, Q, R]
selected = Function('selected', scientist_sort, BoolSort())

pre_conditions = []
```

**ExampleFail:**

Positive example contains undeclared free variables

**Analysis:**

Code incorrect, Example incorrect. Code is correct but does not
implement a constraint that exactly 5 scientists must be selected.
A valid positive example in solver code could not be generated.

**InitialContext:**

At a concert, exactly eight compositions|F, H, L, O, P, R, S, and T|are to be performed exactly once each, consecutively and one composition at a time.

**InitialCode:**

```
from z3 import *

compositions_sort, (F, H, L, O, P, R, S, T) =
EnumSort('compositions', ['F', 'H', 'L', 'O', 'P', 'R', 'S', 'T'])
compositions = [F, H, L, O, P, R, S, T]
positions_sort = IntSort()
position = Function('position', compositions_sort, positions_sort)
pre_conditions = []

pre_conditions.append(Distinct([position(c) for c in
compositions]))
pre_conditions.append(And([position(c) >= 1 for c in
compositions]))
pre_conditions.append(And([position(c) <= 8 for c in
compositions]))
```

**Condition:**

T is performed either immediately before F or immediately after R.

**ConditionCode:**

```
pre_conditions.append(Or(position(T) + 1 == position(F),
position(T) == position(R) + 1))
```

**NegExample:**

```
position(T) == position(F) - 3
```

**NegExampleDescription:**

T is performed with two compositions between it and F.

**Analysis:**

Code correct, Example incorrect. The negative example only requires that T is 3 positions before F, which is permitted by the constraint as long as T is immediately after R (since it was a disjunction).

**InitialContext:**

A government needs to assign new ambassadors to Venezuela, Yemen, and Zambia. The candidates for these ambassadorships are Jaramillo, Kayne, Landon, Novetzke, and Ong. One ambassador will be assigned to each country, and no ambassador will be assigned to more than one country.

**InitialCode:**

```
from z3 import *

candidates_sort, (Jaramillo, Kayne, Landon, Novetzke, Ong) =
EnumSort('candidates', ['Jaramillo', 'Kayne', 'Landon',
'Novetzke', 'Ong'])
countries_sort, (Venezuela, Yemen, Zambia) = EnumSort('countries',
['Venezuela', 'Yemen', 'Zambia'])
candidates = [Jaramillo, Kayne, Landon, Novetzke, Ong]
countries = [Venezuela, Yemen, Zambia]
assigned_to = Function('assigned_to', candidates_sort,
countries_sort, BoolSort())

pre_conditions = []

pre_conditions.append(Distinct([assigned_to(c, country) for c in
candidates for country in countries]))
pre_conditions.append(And([Sum([If(assigned_to(c, country), 1, 0)
for country in countries]) == 1 for c in candidates]))
pre_conditions.append(And([Sum([If(assigned_to(c, country), 1, 0)
for c in candidates]) == 1 for country in countries]))
```

**PosExample:**

```
And(assigned_to(Jaramillo, Venezuela) == True, assigned_to(Kayne,
Yemen) == True, assigned_to(Landon, Zambia) == True,
assigned_to(Novetzke, Venezuela) == False, assigned_to(Novetzke,
Yemen) == False, assigned_to(Novetzke, Zambia) == False,
assigned_to(Ong, Venezuela) == False, assigned_to(Ong, Yemen) ==
False, assigned_to(Ong, Zambia) == False)
```

**PosExampleDescription:**

```
Jaramillo is assigned to Venezuela, Kayne to Yemen, and Landon to
Zambia, while Novetzke and Ong are not assigned to any country.
```

**Analysis:**

```
Code incorrect, Example correct. The positive example correctly
instantiates the problem context, but the code incorrectly
enforces the very strong distinctness constraint on the
assigned_to() function which has boolean return type and should
not have dictinct outputs over all possible assignments.
```

---

**InitialContext:**

```
Of the eight students|George, Helen, Irving, Kyle, Lenore, Nina,
Olivia, and Robert|in a seminar, exactly six will give individual
oral reports during three consecutive days|Monday, Tuesday, and
Wednesday. Exactly two reports will be given each day|one in the
morning and one in the afternoon.
```

**InitialCode:**

```
from z3 import *

students_sort, (George, Helen, Irving, Kyle, Lenore, Nina, Olivia,
Robert) = EnumSort('students', ['George', 'Helen', 'Irving',
'Kyle', 'Lenore', 'Nina', 'Olivia', 'Robert'])
```

```
days_sort, (Monday, Tuesday, Wednesday) = EnumSort('days',
['Monday', 'Tuesday', 'Wednesday'])
time_sort, (morning, afternoon) = EnumSort('time', ['morning',
'afternoon'])
students = [George, Helen, Irving, Kyle, Lenore, Nina, Olivia,
Robert]
days = [Monday, Tuesday, Wednesday]
times = [morning, afternoon]
report = Function('report', students_sort, days_sort, time_sort,
BoolSort())

pre_conditions = []
```

**NegExample:**

```
And(report(George, Monday, morning) == True, report(George,
Tuesday, morning) == True)
```

**NegExampleDescription:**

```
George gives two reports, one on Monday morning and another on
Tuesday morning, which contradicts the constraint that each
student gives only one report.
```

**Analysis:**

```
Code incorrect, Example correct. It does not enforce that a
student cannot give multiple reports, so it will not be possible
for exactly 6 reports to be given by exactly 6 students as
intended. The example correctly violates the constraint by
enforcing George to give two reports in different slots.
```

---

**InitialContext:**

```
A bakery makes exactly three kinds of cookie|oatmeal, peanut
butter, and sugar.
Exactly three batches of each kind of cookie are made each week
(Monday through Friday)
and each batch is made, from start to finish, on a single day.
```

**InitialCode:**

```
from z3 import *

days_sort, (Monday, Tuesday, Wednesday, Thursday, Friday) =
EnumSort('days', ['Monday', 'Tuesday', 'Wednesday', 'Thursday',
'Friday'])
cookie_sort, (oatmeal, peanut_butter, sugar) = EnumSort('cookie',
['oatmeal', 'peanut_butter', 'sugar'])
batch_sort = IntSort()
made_on = Function('made_on', cookie_sort, batch_sort, days_sort)

pre_conditions = []

for cookie in [oatmeal, peanut_butter, sugar]:
    pre_conditions.append(And([Exists([Const('d' + str(i),
    days_sort)], made_on(cookie, i) == Const('d' + str(i),
    days_sort)) for i in range(1, 4)]))
    for day in [Monday, Tuesday, Wednesday, Thursday, Friday]:
```

```
2268          pre_conditions.append(Sum([If(made_on(cookie, i) == day,
2269          1, 0) for i in range(1, 4)]) <= 1)
2270
2271
```

**Condition:**

If no batch of cookies is made on Wednesday, then exactly three batches of cookies are made on Tuesday.

**ConditionCode:**

```
day = Const('day', days_sort)
check_property = Sum([If(made_on(cookie, i) == Tuesday, 1, 0) for
cookie in [oatmeal, peanut_butter, sugar] for i in range(1, 4)])
== 3
pre_conditions.append(check_property)
```

**NegExample:**

```
And(made_on(oatmeal, 1) != Wednesday, made_on(peanut_butter, 1) !=
Wednesday, made_on(sugar, 1) != Wednesday,
made_on(oatmeal, 1) == Tuesday, made_on(peanut_butter, 2) ==
Tuesday, made_on(sugar, 3) != Tuesday)
```

**NegExampleDescription:**

No cookies are made on Wednesday, but only two batches of cookies are made on Tuesday.

**Analysis:**

Code incorrect, Example incorrect. The code directly enforces that exactly three batches are made on Tuesday, regardless of whether any batches are made on Wednesday so it lacks the conditional aspect "if no batch is made on Wednesday" of the intended constraint. The negative example prevents only batch 3 of sugar cookies on tuesday but not other batches, so it is still possible to have three batches on tuesday.

---

**InitialContext:**

An administrator must assign parking spaces to six new employees: Robertson, Souza, Togowa, Vaughn, Xu, and Young. Each of the six employees must be assigned one of the following parking spaces: #1, #2, #3, #4, #5, or #6. No two employees can be assigned the same parking space.

**InitialCode:**

```
from z3 import *

employees_sort, (Robertson, Souza, Togowa, Vaughn, Xu, Young) =
EnumSort('employees', ['Robertson', 'Souza', 'Togowa', 'Vaughn',
'Xu', 'Young'])
parking_spaces_sort = IntSort()
parking_space = Function('parking_space', employees_sort,
parking_spaces_sort)
pre_conditions = []
```

```
2322    pre_conditions.append(Distinct([parking_space(e) for e in
2323    [Robertson, Souza, Togowa, Vaughn, Xu, Young]]))
2324    pre_conditions.append(And([parking_space(e) >= 1 for e in
2325    [Robertson, Souza, Togowa, Vaughn, Xu, Young]]))
2326    pre_conditions.append(And([parking_space(e) <= 6 for e in
2327    [Robertson, Souza, Togowa, Vaughn, Xu, Young]]))
```

**Condition:**

If Young is assigned a higher-numbered parking space than Souza, then Young is assigned parking space #2.

**ConditionCode:**

```
check_property = And(parking_space(Young) > parking_space(Souza),
parking_space(Young) == 2)
pre_conditions.append(check_property)
```

**ExampleFail:**

Positive example not inferred (NONE description)

**Analysis:**

Code incorrect, Example incorrect. The condition implemented just enforces that Young has number 2 and is than Souza, but does not allow for a lower number than Souza as intended. The positive example is not inferred.

---

**InitialContext:**

A bakery makes exactly three kinds of cookie|oatmeal, peanut butter, and sugar. Exactly three batches of each kind of cookie are made each week (Monday through Friday) and each batch is made, from start to finish, on a single day.

**InitialCode:**

```
from z3 import *

days_sort, (Monday, Tuesday, Wednesday, Thursday, Friday) =
EnumSort('days', ['Monday', 'Tuesday', 'Wednesday', 'Thursday',
'Friday'])
cookies_sort, (oatmeal, peanut_butter, sugar) =
EnumSort('cookies', ['oatmeal', 'peanut_butter', 'sugar'])
batch_sort = IntSort()
schedule = Function('schedule', cookies_sort, batch_sort,
days_sort)

pre_conditions = []
```

**NegExample:**

```
And(schedule(oatmeal, 1) == Monday, schedule(oatmeal, 2) ==
Monday, schedule(oatmeal, 3) == Monday, schedule(oatmeal, 4) ==
Monday)
```

**NegExampleDescription:**

Four batches of oatmeal cookies are made on Monday, which contradicts the constraint that exactly three batches of each kind of cookie are made each week.

**Analysis:**

Code incorrect, Example correct. The code does not implement any constraints that exactly 3 batches of each kind of cookie should be made in a week.

---

**InitialContext:**

A government needs to assign new ambassadors to Venezuela, Yemen, and Zambia. The candidates for these ambassadorships are Jaramillo, Kayne, Landon, Novetzke, and Ong. One ambassador will be assigned to each country, and no ambassador will be assigned to more than one country.

**InitialCode:**

```
from z3 import *

ambassadors_sort, (Jaramillo, Kayne, Landon, Novetzke, Ong) =
EnumSort('ambassadors', ['Jaramillo', 'Kayne', 'Landon',
'Novetzke', 'Ong'])
countries_sort, (Venezuela, Yemen, Zambia) = EnumSort('countries',
['Venezuela', 'Yemen', 'Zambia'])
ambassadors = [Jaramillo, Kayne, Landon, Novetzke, Ong]
countries = [Venezuela, Yemen, Zambia]
assigned_to = Function('assigned_to', ambassadors_sort,
countries_sort, BoolSort())

pre_conditions = []
a = Const('a', ambassadors_sort)
c1 = Const('c1', countries_sort)
c2 = Const('c2', countries_sort)
pre_conditions.append(ForAll([a], Sum([If(assigned_to(a, c), 1, 0)
for c in countries]) <= 1))
```

**Condition:**

Either Kayne or Novetzke, but not both, is assigned to one of the ambassadorships.

**ConditionCode:**

```
pre_conditions.append(Xor(Or(assigned_to(Kayne, Venezuela),
assigned_to(Kayne, Yemen), assigned_to(Kayne, Zambia)),
Or(assigned_to(Novetzke, Venezuela), assigned_to(Novetzke, Yemen),
assigned_to(Novetzke, Zambia))))
```

**ExampleFail:**

Positive example contains undeclared free variables

**Analysis:**

Code correct, Example incorrect. The example is not well formed with undeclared free variables, while the code correctly implements the constraint using exclusive OR.

**InitialContext:**

A bakery makes exactly three kinds of cookie|oatmeal, peanut
butter, and sugar. Exactly three batches of each kind of cookie
are made each week (Monday through Friday) and each batch is made,
from start to finish, on a single day.

**InitialCode:**

```
from z3 import *

days_sort, (Monday, Tuesday, Wednesday, Thursday, Friday) =
EnumSort('days', ['Monday', 'Tuesday', 'Wednesday', 'Thursday',
'Friday'])
cookie_sort, (oatmeal, peanut_butter, sugar) = EnumSort('cookie',
['oatmeal', 'peanut_butter', 'sugar'])
batches_sort = IntSort()
make_batch = Function('make_batch', cookie_sort, batches_sort,
days_sort, BoolSort())

pre_conditions = []
days = [Monday, Tuesday, Wednesday, Thursday, Friday]
cookies = [oatmeal, peanut_butter, sugar]

# Constraint: Exactly three batches of each kind of cookie are
made each week
for cookie in cookies:
    pre_conditions.append(Sum([If(make_batch(cookie, i, d), 1, 0)
    for i in range(1, 4) for d in days]) == 3)
```

**NegExample:**

```
And(make_batch(oatmeal, 1, Monday), make_batch(oatmeal, 2,
Tuesday), make_batch(oatmeal, 3, Wednesday), make_batch(oatmeal,
4, Thursday))
```

**NegExampleDescription:**

Four batches of oatmeal cookies are made on Monday, Tuesday,
Wednesday, and Thursday (which contradicts the constraint that
exactly three batches of each kind of cookie are made each week).

**Analysis:**

Code incorrect, Example correct. The code only enforces the
constraint for batch numbers 1,2 and 3, but does not restrict any
other batch numbers from being created on a day. The negative
example uses batch number 4 which is not prevented by the code.

---

**InitialContext:**

A bakery makes exactly three kinds of cookie|oatmeal, peanut
butter, and sugar. Exactly three batches of each kind of cookie
are made each week (Monday through Friday) and each batch is made,
from start to finish, on a single day.

**InitialCode:**

```
2484  from z3 import *
2485
2486  days_sort, (Monday, Tuesday, Wednesday, Thursday, Friday) =
2487  EnumSort('days', ['Monday', 'Tuesday', 'Wednesday', 'Thursday',
2488  'Friday'])
2489  cookies_sort, (oatmeal, peanut_butter, sugar) =
2490  EnumSort('cookies', ['oatmeal', 'peanut_butter', 'sugar'])
2491  batch_sort = IntSort()
2492  make_batch = Function('make_batch', cookies_sort, batch_sort,
2493  days_sort)
2494
2495  pre_conditions = []
2496
```

**NegExample:**

```
2499  And(make_batch(oatmeal, 1) == Monday, make_batch(oatmeal, 2) ==
2500  Monday, make_batch(oatmeal, 3) == Monday, make_batch(oatmeal, 4)
2501  == Monday)
```

**NegExampleDescription:**

Four batches of oatmeal cookies are made on Monday, which contradicts the constraint that exactly three batches of each kind of cookie are made each week.

**Analysis:**

Code incorrect, Example correct. The code does not implement any constraint to require exactly 3 batches of each kind of cookie in a week.

---

**InitialContext:**

Of the eight students|George, Helen, Irving, Kyle, Lenore, Nina, Olivia, and Robert|in a seminar, exactly six will give individual oral reports during three consecutive days|Monday, Tuesday, and Wednesday. Exactly two reports will be given each day|one in the morning and one in the afternoon.

**InitialCode:**

```
from z3 import *

students_sort, (George, Helen, Irving, Kyle, Lenore, Nina, Olivia,
Robert) = EnumSort('students',
                                ['George', 'Helen', 'Irving', 'Kyle',
                                'Lenore', 'Nina', 'Olivia', 'Robert'])
days_sort, (Monday, Tuesday, Wednesday) = EnumSort('days',
['Monday', 'Tuesday', 'Wednesday'])
time_sort, (morning, afternoon) = EnumSort('time', ['morning',
'afternoon'])
students = [George, Helen, Irving, Kyle, Lenore, Nina, Olivia,
Robert]
days = [Monday, Tuesday, Wednesday]
times = [morning, afternoon]
report = Function('report', students_sort, days_sort, time_sort,
BoolSort())
```

```
pre_conditions = []

s = Const('s', students_sort)
d = Const('d', days_sort)
pre_conditions.append(ForAll([s, d], AtMost(report(s, d, morning),
report(s, d, afternoon), 1)))

for day in days:
    pre_conditions.append(Sum([If(report(s, day, morning), 1, 0)
    for s in students]) == 1)
    pre_conditions.append(Sum([If(report(s, day, afternoon), 1, 0)
    for s in students]) == 1)

pre_conditions.append(Sum([If(Or(report(s, d, morning), report(s,
d, afternoon)), 1, 0) for s in students for d in days]) == 6)
```

**Condition:**

```
Helen, Kyle, and Lenore, not necessarily in that order, give the
three morning reports.
```

**ConditionCode:**

```
helen_day = Const('helen_day', days_sort)
kyle_day = Const('kyle_day', days_sort)
lenore_day = Const('lenore_day', days_sort)
pre_conditions.append(And(
    report(Helen, helen_day, morning),
    report(Kyle, kyle_day, morning),
    report(Lenore, lenore_day, morning),
    Distinct([helen_day, kyle_day, lenore_day])
))
```

**NegExample:**

```
And(report(Helen, Monday, morning) == True, report(Kyle, Tuesday,
afternoon) == True)
```

**NegExampleDescription:**

```
Helen gives a report on Monday morning, but Kyle gives his report
in the afternoon.
```

**Analysis:**

```
Code incorrect, Example correct. The code lacks a constraint that
ensures each student gives at most one report during the entire
period,
so it allows Kyle to give a morning and an afternoon report. Since
this should not be allowed, the negative example correctly
violates the requirement
that Kyle gives a morning report by requiring him to give an
afternoon report on Tuesday.
```

---

**InitialContext:**

```
A loading dock consists of exactly six bays numbered 1 through 6
consecutively from one side of the dock to the other.
Each bay is holding a different one of exactly six types of
cargo|fuel, grain, livestock, machinery, produce, or textiles.
```

**InitialCode:**

```
from z3 import *

bays_sort = IntSort()
cargo_sort, (fuel, grain, livestock, machinery, produce, textiles)
= EnumSort('cargo',
                    ['fuel', 'grain', 'livestock', 'machinery',
                    'produce', 'textiles'])
cargo = [fuel, grain, livestock, machinery, produce, textiles]
holding = Function('holding', bays_sort, cargo_sort)

pre_conditions = []

pre_conditions.append(Distinct([holding(b) for b in range(1, 7)]))
pre_conditions.append(And([holding(b) != None for b in range(1,
7)]))
```

**Condition:**

The bay holding livestock has a higher number than the bay holding textiles.

**ConditionCode:**

```
b1 = Const('b1', bays_sort)
b2 = Const('b2', bays_sort)
pre_conditions.append(Exists([b1], And(holding(b1) == livestock,
b1 > 0, ForAll([b2],
                    Implies(holding(b2) == textiles, b1 > b2)))))
```

**NegExample:**

```
And(holding(5) == textiles, holding(1) == livestock)
```

**NegExampleDescription:**

Textiles are in bay 5 and livestock is in bay 1.

**Analysis:**

Code incorrect, Example correct. The variables b1 and b2 in the constraint code representing bay numbers are not properly constrained to be within the valid range of bay numbers (1 through 6). This allows the solver to consider bay numbers outside this range, which leads to the negative example not violating the constraint as expected.

---

**InitialContext:**

A bakery makes exactly three kinds of cookie|oatmeal, peanut butter, and sugar.
Exactly three batches of each kind of cookie are made each week (Monday through Friday) and each batch is made, from start to finish, on a single day.

**InitialCode:**

```
from z3 import *
```

```
days_sort, (Monday, Tuesday, Wednesday, Thursday, Friday) =
EnumSort('days', ['Monday', 'Tuesday', 'Wednesday', 'Thursday',
'Friday'])
cookie_sort, (oatmeal, peanut_butter, sugar) = EnumSort('cookie',
['oatmeal', 'peanut_butter', 'sugar'])
batch_sort = IntSort()
made_on = Function('made_on', cookie_sort, batch_sort, days_sort)

pre_conditions = []

for cookie in [oatmeal, peanut_butter, sugar]:
    pre_conditions.append(Distinct([made_on(cookie, i) for i in
    range(1, 4)]))
```

**Condition:**

Exactly three batches of each kind of cookie are made each week
(Monday through Friday) and each batch is made, from start to
finish, on a single day.

**ConditionCode:**

```
d = Const('d', days_sort)
c = Const('c', cookie_sort)
pre_conditions.append(ForAll([c], Sum([made_on(c, i) == d for i in
range(1, 4) for d in [Monday, Tuesday, Wednesday, Thursday,
Friday]]) == 3))
```

**ExampleFail:**

Positive example contains undeclared free variables

**Analysis:**

Code incorrect, Example incorrect. The code is incorrect as it
only enforces the condition on batch numbers 1..3, but allows any
int values for batch numbers,
so more than 3 batches can potentially be made. The example is
incorrect as it contains undeclared free variables.

---

**InitialContext:**

On each of exactly seven consecutive days (day 1 though day 7), a
pet shop features exactly one of three breeds of kitten|Himalayan,
Manx, Siamese|and exactly one of three breeds of puppy|Greyhound,
Newfoundland, Rottweiler.

**InitialCode:**

```
from z3 import *

days_sort, (day1, day2, day3, day4, day5, day6, day7) =
EnumSort('days', ['day1', 'day2', 'day3', 'day4', 'day5', 'day6',
'day7'])
kitten_breeds_sort, (Himalayan, Manx, Siamese) =
EnumSort('kitten_breeds', ['Himalayan', 'Manx', 'Siamese'])
puppy_breeds_sort, (Greyhound, Newfoundland, Rottweiler) =
EnumSort('puppy_breeds', ['Greyhound', 'Newfoundland',
'Rottweiler'])
days = [day1, day2, day3, day4, day5, day6, day7]
```

```
2700   kitten_breeds = [Himalayan, Manx, Siamese]
2701   puppy_breeds = [Greyhound, Newfoundland, Rottweiler]
2702
2703   featured_kitten = Function('featured_kitten', days_sort,
2704   kitten_breeds_sort)
2705   featured_puppy = Function('featured_puppy', days_sort,
2706   puppy_breeds_sort)
2707
2708   pre_conditions = []
2709
2710   # Add constraints to ensure exactly one breed of kitten and one
2710   breed of puppy is featured each day
2711   d = Const('d', days_sort)
2712   pre_conditions.append(ForAll([d], Sum([If(featured_kitten(d) ==
2713   breed, 1, 0) for breed in kitten_breeds]) == 1))
2714   pre_conditions.append(ForAll([d], Sum([If(featured_puppy(d) ==
2715   breed, 1, 0) for breed in puppy_breeds]) == 1))
2716
```

**Condition:**

If Himalayans are not featured on day 7, then day 1 and day 3
CANNOT feature both the same breed of kitten and the same breed of
puppy.

**ConditionCode:**

```
check_property = And(featured_kitten(day1) ==
featured_kitten(day3), featured_puppy(day1) ==
featured_puppy(day3), featured_kitten(day7) != Himalayan)
pre_conditions.append(check_property)
```

**PosExample:**

```
And(featured_kitten(day7) != Himalayan, featured_kitten(day1) ==
Siamese, featured_puppy(day1) == Greyhound, featured_kitten(day3)
== Manx, featured_puppy(day3) == Newfoundland)
```

**PosExampleDescription:**

Himalayans are not featured on day 7, and day 1 features a Siamese
kitten and a Greyhound puppy while day 3 features a Manx kitten
and a Newfoundland puppy.

**Analysis:**

Code incorrect, Example correct. The code implements the condition
incorrectly by just enforcing a conjunction of constraints rather
than the conditional requirement.

---

**InitialContext:**

A bakery makes exactly three kinds of cookie|oatmeal, peanut
butter, and sugar. Exactly three batches of each kind of cookie
are made each week (Monday through Friday) and each batch is made,
from start to finish, on a single day.

**InitialCode:**

```
from z3 import *
```

```
days_sort, (Monday, Tuesday, Wednesday, Thursday, Friday) =
EnumSort('days', ['Monday', 'Tuesday', 'Wednesday', 'Thursday',
'Friday'])
cookie_sort, (oatmeal, peanut_butter, sugar) = EnumSort('cookie',
['oatmeal', 'peanut_butter', 'sugar'])
batch_sort = IntSort()
made_on = Function('made_on', cookie_sort, batch_sort, days_sort)

pre_conditions = []

# Add constraints to ensure exactly three batches of each kind of
cookie are made each week
for cookie in [oatmeal, peanut_butter, sugar]:
    i = Const('i', batch_sort)
    j = Const('j', batch_sort)
    pre_conditions.append(Sum([If(made_on(cookie, k) == day, 1, 0)
    for k in range(1, 4) for day in [Monday, Tuesday, Wednesday,
    Thursday, Friday]]) == 3)
    pre_conditions.append(And([Not(Exists([i, j], And(i != j,
    made_on(cookie, i) == made_on(cookie, j)))) for day in
    [Monday, Tuesday, Wednesday, Thursday, Friday]]))
```

**PosExample:**

```
And(made_on(oatmeal, 1) == Monday, made_on(oatmeal, 2) ==
Wednesday, made_on(oatmeal, 3) == Friday,
made_on(peanut_butter, 1) == Tuesday, made_on(peanut_butter, 2) ==
Thursday, made_on(peanut_butter, 3) == Friday,
made_on(sugar, 1) == Monday, made_on(sugar, 2) == Tuesday,
made_on(sugar, 3) == Thursday)
```

**PosExampleDescription:**

Three batches of oatmeal cookies are made on Monday, Wednesday,
and Friday. Three batches of peanut butter cookies are made on
Tuesday, Thursday, and Friday. Three batches of sugar cookies are
made on Monday, Tuesday, and Thursday.

**Analysis:**

Code incorrect, Example correct. The code incorrectly enforces the
constraint that a cookie cannot be made on more than one day
(which is not intended).

---

**InitialContext:**

A bakery makes exactly three kinds of cookie|oatmeal, peanut
butter, and sugar. Exactly three batches of each kind of cookie
are made each week (Monday through Friday) and each batch is made,
from start to finish, on a single day.

**InitialCode:**

```
from z3 import *

days_sort, (Monday, Tuesday, Wednesday, Thursday, Friday) =
EnumSort('days', ['Monday', 'Tuesday', 'Wednesday', 'Thursday',
'Friday'])
cookie_sort, (oatmeal, peanut_butter, sugar) = EnumSort('cookie',
['oatmeal', 'peanut_butter', 'sugar'])
```

```
batch_sort = IntSort()
made_on = Function('made_on', cookie_sort, batch_sort, days_sort)

pre_conditions = []
```

**NegExample:**

```
And(made_on(oatmeal, 1) == Monday, made_on(oatmeal, 2) == Tuesday,
made_on(oatmeal, 3) == Wednesday, made_on(oatmeal, 4) == Thursday)
```

**NegExampleDescription:**

Four batches of oatmeal cookies are made, which contradicts the constraint that exactly three batches of each kind of cookie are made each week.

**Analysis:**

Code incorrect, Example correct. The code does not implement any constraint that enforces that exactly 3 batches of each type of cookie should be made in the week.

---

**InitialContext:**

A bakery makes exactly three kinds of cookie|oatmeal, peanut butter, and sugar. Exactly three batches of each kind of cookie are made each week (Monday through Friday) and each batch is made, from start to finish, on a single day.

**InitialCode:**

```
from z3 import *

days_sort, (Monday, Tuesday, Wednesday, Thursday, Friday) =
EnumSort('days', ['Monday', 'Tuesday', 'Wednesday', 'Thursday',
'Friday'])
cookies_sort, (oatmeal, peanut_butter, sugar) =
EnumSort('cookies', ['oatmeal', 'peanut_butter', 'sugar'])
batch_sort = IntSort()
schedule = Function('schedule', cookies_sort, batch_sort,
days_sort)

pre_conditions = []

# Constraint: Exactly three batches of each kind of cookie are
made each week
for cookie in [oatmeal, peanut_butter, sugar]:
    pre_conditions.append(Sum([If(schedule(cookie, i) == day, 1,
    0) for i in range(1, 4) for day in [Monday, Tuesday,
    Wednesday, Thursday, Friday]]) == 3)
```

**NegExample:**

```
And(schedule(oatmeal, 1) == Monday, schedule(oatmeal, 2) ==
Tuesday, schedule(oatmeal, 3) == Wednesday, schedule(oatmeal, 4)
== Thursday)
```

**NegExampleDescription:**

Four batches of oatmeal cookies are made on Monday, Tuesday, Wednesday, and Thursday (which contradicts the constraint that exactly three batches of each kind of cookie are made each week).

**Analysis:**

```
Code incorrect, Example correct. The code only implements the
constraint for batch numbers 1, 2, and 3, but not for any other
batch numbers.
```

---

**InitialContext:**

```
On each of exactly seven consecutive days (day 1 though day 7), a
pet shop features exactly one of three breeds of kitten|Himalayan,
Manx, Siamese|and exactly one of three breeds of puppy|Greyhound,
Newfoundland, Rottweiler.
```

**InitialCode:**

```
from z3 import *

days_sort = IntSort()
kitten_breeds_sort, (Himalayan, Manx, Siamese) =
EnumSort('kitten_breeds', ['Himalayan', 'Manx', 'Siamese'])
puppy_breeds_sort, (Greyhound, Newfoundland, Rottweiler) =
EnumSort('puppy_breeds', ['Greyhound', 'Newfoundland',
'Rottweiler'])

featured_kitten = Function('featured_kitten', days_sort,
kitten_breeds_sort)
featured_puppy = Function('featured_puppy', days_sort,
puppy_breeds_sort)

pre_conditions = []

pre_conditions.append(And([featured_kitten(i) != featured_kitten(i
+ 1) for i in range(1, 7)]))
pre_conditions.append(And([featured_puppy(i) != featured_puppy(i +
1) for i in range(1, 7)]))
```

**Condition:**

```
No breed is featured on any two consecutive days.
```

**ConditionCode:**

```
# (Already implemented above)
```

**ExampleFail:**

```
Positive example not inferred (NONE description)
```

**Analysis:**

```
Code correct, Example incorrect. No example was inferred. The code
correctly implements constraints to disallow the same breeds on
any two consecutive days.
```

---

**InitialContext:**

An administrator must assign parking spaces to six new employees:
Robertson, Souza, Togowa, Vaughn, Xu, and Young. Each of the six
employees must be assigned one of the following parking spaces:
#1, #2, #3, #4, #5, or #6. No two employees can be assigned the
same parking space.

**InitialCode:**

```
from z3 import *

employees_sort, (Robertson, Souza, Togowa, Vaughn, Xu, Young) =
EnumSort('employees', ['Robertson', 'Souza', 'Togowa', 'Vaughn',
'Xu', 'Young'])
parking_spaces_sort = IntSort()
parking_space = Function('parking_space', employees_sort,
parking_spaces_sort)

pre_conditions = []

pre_conditions.append(Distinct([parking_space(e) for e in
[Robertson, Souza, Togowa, Vaughn, Xu, Young]]))
pre_conditions.append(And([parking_space(e) >= 1 for e in
[Robertson, Souza, Togowa, Vaughn, Xu, Young]]))
pre_conditions.append(And([parking_space(e) <= 6 for e in
[Robertson, Souza, Togowa, Vaughn, Xu, Young]]))
```

**Condition:**

If Young is assigned a higher-numbered parking space than Souza,
then Young is assigned parking space #2.

**ConditionCode:**

```
check_property = parking_space(Young) == 2
pre_conditions.append(check_property)
```

**ExampleFail:**

Positive example not inferred (NONE description)

**Analysis:**

Code incorrect, Example incorrect. The code does not implement the
conditional requirement but simply asserts that Young should be
assigned #2. No positive example is inferred.

---

**InitialContext:**

A bakery makes exactly three kinds of cookie|oatmeal, peanut
butter, and sugar. Exactly three batches of each kind of cookie
are made each week (Monday through Friday) and each batch is made,
from start to finish, on a single day.

**InitialCode:**

```
from z3 import *

cookies_sort, (oatmeal, peanut_butter, sugar) =
EnumSort('cookies', ['oatmeal', 'peanut_butter', 'sugar'])
```

```
days_sort, (Monday, Tuesday, Wednesday, Thursday, Friday) =
EnumSort('days', ['Monday', 'Tuesday', 'Wednesday', 'Thursday',
'Friday'])
batches_sort = IntSort()
made_on = Function('made_on', cookies_sort, batches_sort,
days_sort)

pre_conditions = []

# Ensure that each kind of cookie has exactly three batches made
on different days
for cookie in [oatmeal, peanut_butter, sugar]:
    pre_conditions.append(Distinct([made_on(cookie, b) for b in
    range(1, 4)]))

# Ensure that each batch number is between 1 and 3
for cookie in [oatmeal, peanut_butter, sugar]:
    for b in range(1, 4):
        pre_conditions.append(And(b >= 1, b <= 3))
```

**NegExample:**

```
And(made_on(oatmeal, 1) == Monday, made_on(oatmeal, 2) == Tuesday,
made_on(oatmeal, 3) == Wednesday)
```

**NegExampleDescription:**

```
The oatmeal cookies are made on Monday, Tuesday, and Wednesday,
which violates the constraint that each kind of cookie must be
made on different days.
```

**Analysis:**

```
Code incorrect, Example incorrect. The negative example does not
violate the intended constraints as it simply assigns oatmeal
batches to 3 different days. The code does not prevent any batch
numbers higher than 3.
```

A.6    RUNTIME PERFORMANCE AND OPTIMIZATIONS

We conducted an evaluation of the runtime performance of the current system. Executing the system over a sample of 250 data points (50 from each dataset), the median runtime per task is 152 seconds (around 2.5 minutes), with first quartile 108s, third quartile 267s and mean 249s. This was on an Intel Xeon Gold 6126 CPU @ 2.60 GHz with 16 cores and no hyper-threading, 62 GB of RAM, and an HDD-based storage system (this machine has slightly lower single-threaded performance than most modern desktops). However, there are also many potential optimizations to the SSV algorithm that can be made to significantly reduce the run time in a practical implementation:

- The outer temperature loop (line 2 in Figure 4) can be fully parallelized as all the computations are independent for each temperature. That can yield up to 4X speed up (with 4 temperatures being tried in our current system). Side note: even with a single temperature of 0, our algorithm still beats all baselines in terms of accuracy (as in our ablation study), so even such an ablated system would be beneficial if computation costs are of significant concern.

- In the verification phase (line 9 in Figure 4), the solver calls to verify each of the concrete instantiations can be parallelized as they are checked independently. These are around 10 to 20 independent solver calls on average (2 instantiations each for around 5-10 constraints) that can be parallelized for significant speedup.

- Caching solver verification checks between repair attempts. Currently for each repair attempt in the inner loop (line 4 in Figure 4), we perform the full verification on the repaired

program (on all constraints). However, most of the time the repaired change is on a single constraint for which a failing instantiation was found and all other constraints remain identical (though not always guaranteed as in some rare cases the LLM may reformulate the whole program). Hence if we cache the solver requests for each instantiation verification, many of these repetitive checks can be avoided in the repaired programs for the constraints that are unaltered.

As a general side note, recent reasoning-oriented models such as Open AI's o1 can take several seconds or up to a few minutes on some tasks with significantly more computational resources/GPUs, so higher runtimes in the order of a few minutes may generally be expected to robustly address complex reasoning problems.

