# OpenReview forum: "Almost Sure Reasoning: Generating Verified Formalizations with Language Models and Logical Solvers"
_ICLR.cc/2025/Conference — Submitted to ICLR 2025_

### Official Review · Reviewer_GMco · 2024-11-04

**Soundness:** 4
**Presentation:** 3
**Contribution:** 2
**Rating:** 6
**Confidence:** 3

**Summary:**

This paper considers the problem of autoformalization of natural language queries for use in with formal methods, e.g. automata theorem provers and SMT solvers. This work extends existing works on proposer/verifier LLM architectures by having the LLM generate both the autoformalization and the concrete instantiations (unit tests). These tests act to guide to correct the autoformalization. The paper then validates the approach and empirically finds this to be a successful addition to the LLM toolbox.

**Strengths:**

1. The addressed problem of autoformalization is interesting and the potential benefits are clear: improved reasoning and synthesis using formal methods.

2. The presentation is clear and easy to follow. The core idea is summarized and illustrated well.

3. The two key ideas of adding concrete instantiations to check against and adding a feedback signal of whether the answer was verified are simple and applicable to a wide number of domains. In particular, while simple, this idea does differentiate itself from a number of AI systems papers targeting autoformalization via the proposer/verifier pattern.

4, The empirical benchmarks show a clear improvement over the baselines.

5. The limitations does a good job of outlining the limits, particularly in problems that cannot be identified with a finite number of concrete instances.

**Weaknesses:**

1. The approach appears to be a simple extension of the proposer/verifier pattern -- Although I think many great ideas can be small / surgical modifications to existing approaches / frameworks.

2. Perhaps I'm missing something, but it's unclear from the draft how many concrete instantiations are necessary or if tuning this is important. For example, I can imagine a trade off where too many might increase the likelihood of an incorrect example slipping in. Conversely too few do not exercise edge cases.

3. (minor) I must admit I do not like the language around almost sure reasoning. To me this sounds to much like "almost surely" "almost certainly", etc from probability theory. These have a precise meaning that if I understand the paper, this approach does not and can never have.

**Questions:**

1. How many instances were generated per round.

2. In the limit of many instances, does it make sense to benchmark against passive learning algorithms for the concepts being tested?

---

> ### Author Response · Authors · 2024-11-19
> **Response to Reviewer GMco [1/2]**
>
> Thank you to the reviewer for the very helpful comments.
>
> **The approach appears to be a simple extension of the proposer/verifier pattern -- Although I think many great ideas can be small / surgical modifications to existing approaches / frameworks.**
>
> [Please also see our response to reviewer 9aEs for more details of our conceptual novelty]. Standard proposer-verifier approaches either depend on a formal (sound) verifier that can reliably check correctness for the proposer model, or an informal/uncertain verifier (e.g. an LLM self-critiquing) that may make relative improvements to overall accuracy. In our case, as we are inferring a formalization from an informal input problem, there does not exist any formal verifier to check correctness. Instead, the novelty in our approach is that the proposer proposes both the solution (the formalization) and the test cases and the verifier only checks *consistency* between them (rather than *correctness* of the solution). We show how this consistency-based verification yields near perfect empirical precision in our case even though it cannot always guarantee correctness due to the informal nature of our problem. Hence the key distinctions:
>
> - we do not have a formal verifier to check correctness like prior approaches, but can ensure correctness with high precision using our consistency-based verification of abstract and concrete proposals
> - prior approaches that use informal/uncertain verifiers can only provide relative accuracy improvements, while our consistency-based formal verification can provide near-perfect precision - this is why our approach can actually offer verification as an additional standalone feature of our system (which prior approaches could not).
>
> **It's unclear from the draft how many concrete instantiations are necessary or if tuning this is important. For example, I can imagine a trade off where too many might increase the likelihood of an incorrect example slipping in.**
>
> In our current implementation, we infer one positive and one negative example instantiation per constraint, and there are around 5 – 10 constraints inferred in a given question. Hence there are around 10 to 20 instantiations inferred for each formalization (program) that is inferred for a problem. This strategy was fixed in our current implementation, but the reviewer is correct that variations in the number of instantiations would be interesting to explore. As discussed in our limitations section, we found a very small number of cases with GPT3.5 where more instantiations may have helped, and this would be good to explore in future work. You also correctly point out that there is trade-off in that more instantiations increase the likelihood of inferring incorrect ones. This is expected to affect the coverage of verification rather than precision - since verification is based on agreement between the formalization and the instantiations and hence incorrect instantiations are unlikely to pass verification – even in the current implementation we infer some incorrect instantiations but these cause verification failures rather than false positives, thus maintaining high precision (see response to reviewer “9aEs” where we find 10% of sample verification failures are due to wrong instantiations). In general, the amount of instantiations would be interesting to experiment with depending on target application domains or preference for precision vs coverage. We also note that our compositional approach of inferring instantiations for specific independent constraints in the problem (rather than the whole problem) also helps in comprehensive verification of different aspects of the problem.
>
> **I do not like the language around almost sure reasoning.**
>
> That is a good point. We agree it sounds too similar to those notions in probability theory and does not mean the same thing, which can be confusing. We wanted to coin a phrase that qualitatively distinguishes the approach/paradigm of providing extremely high empirical precision of verification (whether it is our specific SSV technique or any others in future). In contrast to traditional approaches where as benchmark complexity increases the overall system accuracy goes down, with “almost sure reasoning” approaches, as benchmark complexity increases, the precision should remain very high though coverage of verification may go down - and even zero if the benchmarks become too hard for the system. So such techniques are “almost sure” when they do verify, though they may not always verify. We wanted to have a phrase that can identify/distinguish such approaches to reasoning, as progress in this direction is practically important to achieve reliability/autonomy of AI systems by reducing manual verification burden in at least some proportion of cases in practice. Perhaps another term like “nearly sure reasoning” or “near-perfect reasoning”  would be better? We are also open to suggestions and can change the phrasing in the paper and title.

---

> ### Author Response · Authors · 2024-11-19
> **Response to Reviewer GMco [2/2]**
>
> **How many instances were generated per round.**
>
> We infer one positive and one negative example instantiation per constraint, and there are around 5 – 10 constraints inferred in a given question. Hence there are around 10 to 20 instantiations inferred for each formalization (program) that is inferred for a problem. And of course multiple formalizations are inferred in the algorithm over different repair attempts and different temperatures (we try 4 different temperatures and up to 4 repair attempts for each temperature).
>
>
> **In the limit of many instances, does it make sense to benchmark against passive learning algorithms for the concepts being tested?**
>
> We are not exactly sure if we understand the question correctly, and feel free to provide further clarification if we have misinterpreted. In the limit of many instances one can imagine that for certain tasks, a passive learning algorithm such as a specialized classifier may be able to generate or verify examples much better than a general purpose LLM. We could hence consider benchmarking against using such an algorithm for examples generation in place of the LLM in our SSV approach for those kinds of tasks. However, it is difficult to see how this can be generally applied for arbitrary reasoning problems. If the question is asking whether we can compare the entire SSV approach with passive learning algorithms, then that is less clear to us, as it is difficult to see how a black box learning algorithm can reliably infer the answer to a new reasoning problem based purely on many sample instantiations of past problems (it can certainly be tried though). We apologize if we are misunderstanding the question and please let us know if you meant something different.

---

> ### Author Response · Authors · 2024-11-25
>
> We have made further revisions to the paper to add the above points about our conceptual novelty in comparison with standard proposer-verifier approaches in the introduction. We have also replaced all occurrences of the phrase "almost sure" with "near-certain". We hope these address the reviewer's concerns.

---

> > ### Comment · Reviewer_GMco · 2024-11-27
> >
> > Thank you for the detailed responses to my review. Having read the other reviews and the discussion, I would like to maintain my current thinking that this is a borderline accept paper.
> >
> > I see the key novelty in checking consistency and appreciate the change of almost sure to near-certain.
> >
> > Regarding the question of benchmarking against passive algorithms, this only made sense if you were generating many concrete instances. As you clarified, only a few instances are created, so it doesn't really make sense.

---

> > > ### Author Response · Authors · 2024-11-27
> > >
> > > Thank you to the reviewer for your very valuable feedback which has helped to improve the paper.

---

### Official Review · Reviewer_hoJe · 2024-11-04

**Soundness:** 2
**Presentation:** 2
**Contribution:** 1
**Rating:** 3
**Confidence:** 2

**Summary:**

This paper introduces Semantic Self-Verification (SSV), a method to combine large language models (LLMs) with logical solvers. Specifically, the LLM is used to infer a formal representation of the problem that can then be solved by the solver.

**Strengths:**

The main contributions of Semantic Self-Verification (SSV) are summarized by the authors as follows:

1) It improves the overall accuracy of reasoning significantly over SoTA.
2) It provides a novel feature of verification that has near-perfect precision.

**Weaknesses:**

1) The introduction is 4 pages, which is perhaps too much for a paper which is 10 pages in total.
For instance, on p. 3 there is a long example, which is hard to appreciate if one doesn't already know the details of the proposed method.

2) The discussion in the paper is always rather informal, so it often hard to understand what is going one. For instance, on p. 5 the authors say that "our program generation phase creates programs with an explicit structure P_init + C_1... + C_N + O_1 + ... + O_M", where P_init is the initial definitions segment, and the Cs and Os are constraints and options. However, the form of these constraints and options is not specified.

3) Related to the previous point, it is not clear to which kind of problems we can apply the method proposed in this submission. Are the authors claiming that their methods applies to any problem that can be passed to an LLM? The limitations (if any) of SSV are never really discussed in the paper.

4) Overall the contribution seems a bit generic and limited to the SSV algorithm. But basically any LLMs can be considered, any benchmark, and possibly any solver.
One is left wondering what is the actual contribution of the paper.

**Questions:**

I'd ask the authors to elaborate and possibly reply to weaknesses (3) and (4) above.

---

> ### Author Response · Authors · 2024-11-19
> **Response to Reviewer hoJe**
>
> Thank you to the reviewer for the very helpful comments.
>
> **Long introduction**
>
> This is a good point. It would be better to have shorter high-level introduction and then a separate section that outlines the approach in more detail using the running/motivating example. We will make this improvement.
>
> **Discussion informal. On p. 5 the authors say that "our program generation phase creates programs with an explicit structure P_init + C_1... + C_N + O_1 + ... + O_M", where P_init is the initial definitions segment, and the Cs and Os are constraints and options. However, the form of these constraints and options is not specified.**
>
> We will work on improving the formalization and clarify the presentation better. In this particular case, each of P_init, C_i and O_i are segments of code in the language of the solver (in our particular implementation the z3 solver). For example, in Figure 1b, P_init is all of the code that has initial declarations up to the first constraint comment “#CONSTRAINT:”, while C_1 is all of the code between this comment and the next constraint comment, and so on. This code segment and comments structure is specified in our code generation prompts to the LLM so that the code implementing each constraint can be easily extracted for verification.
>
> **It is not clear to which kind of problems we can apply the method proposed in this submission.**
>
> Yes we will work on more clearly describing the scope of problems we are addressing. The method is designed to address any problems that require logical or deductive reasoning: where there are some premises (constraints) given in the question and the answer should follow from the premises based on pure logical deduction. Such tasks present a difficult challenge for LLMs as they often make incorrect deductions in long chains of inference, as illustrated by the reasoning benchmarks that we target here. More concretely, our approach should address any problem that can possibly be expressed in a formal logical language and where there exists a corresponding solver (e.g. general First Order Logic that is expressible in the Z3 solver that we use here). Our method is not meant for non-reasoning tasks e.g. simple question answering, information retrieval, generative tasks such as poetry/art/content generation, sentiment analysis etc. However, please also see the discussion with reviewer “SW9F” on how the scope of our SSV technique can potentially be expanded beyond pure deductive inference to more uncertain abductive/inductive inference by inferring likely background knowledge from LLMs.
>
> **Overall the contribution seems a bit generic and limited to the SSV algorithm - what is the actual contribution of the paper.**
>
> Our core contribution is this: we present a new AI system that can address purely deductive logical reasoning tasks much better than the state-of-the art systems in two important respects:
>
> - It has significantly higher accuracy than prior reported systems on standard deductive reasoning benchmarks
> - It has a novel feature of verification where it can indicate in which cases it has almost sure confidence in its answer (and this precision of verification is 100% in our benchmarks - meaning that the system is always correct when it signals that it has verified a case). Such a verification feature does not exist in any prior reasoning systems which just answer all questions without any confidence indication (and get many of them wrong).
>
> Beyond this core concrete contribution, our system is based on a novel technique that incorporates LLMs and symbolic solvers in a new way (the SSV algorithm), and this algorithm is parametric in both the LLM and the solver. The key conceptual novelty in this algorithm is to use a proposer-verifier approach where, unlike standard approaches where the proposer proposes a solution and the verifier can check *correctness* of the solution, in our case the proposer proposes both a solution (a formalization) and test cases and the verifier only checks *consistency* between them (since there is no verifier that can possibly check correctness for our informal-to-formal prediction task). And we show how such formal consistency-based verification can yield near-perfect precision empirically [Please also see the discussion with reviewer 9aEs for more details on our conceptual novelty]
>
> While our SSV algorithm is parametric in both the LLM and the solver and we have tested it with two mainstream LLMs and a state of the art solver, we do not make further claims about its generalization to arbitrary models or solvers as a key contribution – but only observe that the method is generalizable in this respect by design and can hence potentially be explored in this dimension.

---

> ### Author Response · Authors · 2024-11-25
>
> We have made further revisions to the paper to separate the introduction into a smaller section and added a new  section that describes the motivating example in more detail. We have also clarified our problem scope to deductive reasoning tasks and our conceptual novelty in comparison with standard proposer-verifier approaches in the introduction.

---

> > ### Comment · Reviewer_hoJe · 2024-11-26
> >
> > I thank the authors for the time they have taken to address my concerns.
> >
> > Unfortunately, the answers provided by the authors confirm my fears on the generic scope of the paper. The authors say that "[t]he method is designed to address any problems that require logical or deductive reasoning."
> > To back up such a statement the authors should provide an in-depth experimental evaluation across several different deductive tasks, which doesn't seem the case.
> >
> > In my opinion the paper is not significant enough for a prestigious conference such as ICLR.

---

> > > ### Author Response · Authors · 2024-11-27
> > >
> > > Thank you to the reviewer for the additional comment.
> > >
> > > In terms of significance, the topic of combining LLMs and logical solvers has appeared in top recent conferences such as NeurIPS 2023 (Ye et al) and EMNLP 2023 (Pan et al). The limits of both these works highlight the key open challenge which is to ensure that the formalization is correct with respect to the NL description, and ours is the first work that addresses this challenge to advance the field towards formal reasoning using LLMs.
> > >
> > > Many ICLR papers have also focused on the generic scope of reasoning with LLMs (Saparov et al 2023, Yu et al 2024, Creswell et al 2023, Wang et al 2023 ...) and have used many of the same and a similar number of benchmarks as we have to demonstrate value. In fact, we have used *all* of the same standard reasoning benchmarks that were used in the closest related SoTA work on this topic (Pan et al EMNLP'23), and show significant gains both in accuracy and our novel verification feature that no work on reasoning has previously provided. This is especially highlighted on the real AR-LSAT law school tests dataset where we achieve 71% accuracy as compared to all prior works where accuracy has been up to 43%, and our novel verification feature additionally provides 100% correctness guarantee on 21% of cases.

---

> > > > ### Comment · Reviewer_hoJe · 2024-11-28
> > > >
> > > > Hi,
> > > >
> > > > I'm sorry for being blunt, but it seems to me that the paper just combine LLMs and solvers, where LLMs are used to provide a formalisation which is then passed to the solver.
> > > >
> > > > I might be reductive, but I fail to see what is the challenge and the innovation of basically combining already existing blocks in a rather straightforward manner.
> > > >
> > > > Happy to discuss the matter further.

---

> > > > > ### Author Response · Authors · 2024-11-28
> > > > >
> > > > > We would like to thank the reviewer for their continued interest and allowing us the opportunity to clarify our contribution.
> > > > >
> > > > > Actually, what you say about "LLMs are used to provide a formalisation which is then passed to the solver" is what is done in prior work (Pan et al EMNLP'23 and Ye et al NeurIPs'23). But the key challenge that is highlighted by the limits of these works is that the LLMs can easily produce **the wrong formalization** of the problem (e.g. using a wrong quantifier or operator somewhere) and there is no way to know if a given formalization actually represents the original NL problem correctly (even one small mistake somewhere can cause **the wrong problem to be solved** by the solver). This is concretely evident in that the best reported accuracy of prior work on the real law school tests dataset is only 43% (so 57% of the time the LLM produces the wrong formalization and the solver solves the wrong problem). So how can a system know if it is solving *the right* problem? There is no formal way to check that since the input problem is informally stated in NL. And the model always produces *some* formalization for any problem it is given - how can the system know when the arbitrary code it produces is correct and when it isnt? This is an open challenge of autoformalization that also appears in many related domains like planning, math proofs, etc.
> > > > >
> > > > > To address this central challenge, the key novel idea in our work is that we use the LLM to not just produce the abstract formalization, but also generate concrete instances of the NL problem, and we use the solver to check **consistency between the abstract and concrete inferences**. The underlying concept is that the abstract and concrete inferences are unlikely to be consistent with each other if the LLM is misinterpreting the problem  - even though each of them can often be wrong, when there is agreement between them then we have a near-certain confidence in the correctness. This provides our very strong novel notion of verification that **no prior work on reasoning has been able to provide** - we have a system that can state when it is sure of the answer and when it is, it is **always correct!** (i.e. empirically we show verification has 100% precision on our benchmarks). No prior reasoning approach (whether solver-based or not) has been able to provide this kind of a verification feature. Having this near-certainty can be hugely beneficial in practice, as we can avoid manually checking the answer in the cases where the system signals it is sure.
> > > > >
> > > > > Moreover, not only do we provide this new notion of near-certain verification, but we also show how it can be used to further improve the formalizations themselves - using failing verification examples to improve formalizations, and also exploring multiple formalizations and choosing the ones where our verification succeeds. This allows us to achieve a huge boost of 71% accuracy on the law school benchmark as compared to the 43% of the SoTA approaches (and also on other benchmarks).
> > > > >
> > > > > Hence to summarize, below are the most important contributions of this work:
> > > > >
> > > > > - We provide a novel feature of verification where the system indicates with near-perfect certainty if it is correct (100% correctness on our benchmarks) - no prior work on reasoning has been able to offer such a feature.
> > > > > - Empirically, we significantly boost the answer accuracy on reasoning tasks over SoTA (over all benchmarks, and in particular from 43% to 71% on the most difficult law school tests benchmark)
> > > > > - Conceptually, our technique brings a new novelty to proposer-verifier approaches: in contrast to the standard paradignm that uses verifiers to check *correctness* in formal problem tasks, we show how one can leverage the notion of *consistency* between abstract and concrete inferences to use formal verifiers for informal-to-formal prediction tasks - with near-certain correctness (as full correctness can never be guaranteed for such tasks).

---

### Official Review · Reviewer_9aEs · 2024-11-05

**Soundness:** 3
**Presentation:** 4
**Contribution:** 2
**Rating:** 5
**Confidence:** 4

**Summary:**

The paper presents a new method called Semantic Self-Verification (SSV), designed to improve the reliability of AI reasoning by combining LLMs with logical solvers for semantic in-context feedback. Previous AI reasoning often relies solely on LLMs, which can make mistakes, especially on complex problems. SSV addresses this by having the LLM translate a problem into a formal logical representation, then using a logical solver to verify that the translation accurately reflects the problem's constraints.

SSV does this by generating concrete instantiations or specific examples, that test each rule or constraint in the problem. For instance, if a rule says "Stacy and Yolanda cannot repair the same machines," SSV will create scenarios where this rule is both upheld and violated, then use the solver to confirm that the rule is consistently applied across these cases. If an answer option satisfies all constraints under these test cases, SSV flags it as "verified". In experiments on complex reasoning benchmarks, such as those from law and logic exams, SSV achieved significantly higher accuracy than existing approaches. This method not only improves accuracy but also helps reduce the need for human intervention in verifying answers, especially on high-confidence cases.

**Strengths:**

1. SSV introduces an interesting approach by generating concrete examples to verify the logical accuracy of language model outputs, rather than relying on the model’s abstract reasoning.
2. The method achieves near-perfect precision on verified cases, meaning it is highly reliable for a subset of answers.

**Weaknesses:**

1. My main concern is the novelty aspect. Its main innovation is the concrete instantiation, that seems like an incremental improvement on the established method of combining LLMs with logical solvers. The use of counter-example guided feedback, in which semantic verification is conducted through a solver, has been explored in other reasoning tasks, such as in Stechly et al. [1], which examines the self-verification limitations of LLMs in reasoning and planning, and in Jia et al. [2], which explores counter-example guided feedback in material discovery tasks.
2. SSV’s success relies heavily on the LLM's ability to produce accurate formal representations and concrete instantiations, which can vary in quality, particularly on ambiguous tasks. This has also been addressed in the limitations section, but it still remains an important concern.
3. While SSV is precise in verified cases, the coverage is relatively low on challenging datasets (e.g., AR-LSAT). This limited coverage raises questions about the method's practicality, as it may leave many complex cases unverifiable.
4. The paper could improve by providing a more detailed analysis of cases where SSV fails verification. Understanding common sources of error could help readers evaluate the limitations more clearly.

Minor formatting issues:
1. Latex formatting issues: Use `` for opening quotes - ``Gary is nice.”
2. For references, write “we use the same datasets as in Pan et al. (2023)” and apply ~\cite{} instead of ~\citep{} for such cases.

[1] Stechly, Kaya, Karthik Valmeekam, and Subbarao Kambhampati. "On the self-verification limitations of large language models on reasoning and planning tasks." arXiv preprint arXiv:2402.08115 (2024).
[2] Jia, Shuyi, Chao Zhang, and Victor Fung. "LLMatDesign: Autonomous Materials Discovery with Large Language Models." arXiv preprint arXiv:2406.13163 (2024).

**Questions:**

The paper covers several limitations, such as the difficulty of verifying natural language accurately and how errors might occur if examples are not comprehensive. Moreover, SSV may struggle with large, unstructured data, high computational costs, and performance variations with different language models. Additionally, its ability to verify complex or ambiguous language is limited, and its effectiveness outside formal logic tasks is still uncertain.

It would be great if the authors can address the concerns raised above as well as in the Weaknesses section.

---

> ### Author Response · Authors · 2024-11-19
> **Response to Reviewer 9aEs [1/4]**
>
> Thank you to the reviewer for the very helpful comments.
>
> **Novelty**
>
> It is indeed important point to clearly convey the conceptual novelty of our technique, which we believe is significant (apart from our strong empirical results and the new application domain). The novelty of our technique is not just about concrete instantiations (which the reviewer rightly points out have been used in many works) but in the different way in which they are used in our informal-to-formal setting. In prior settings that reviewer mentions, there is commonly a *formal* **problem** specified, a **proposer** model that proposes a solution and a formal **verifier** that can reliably check the correctness of the solution and provide counterexamples/feedback. For example, in Stechly et al. [1] there is a formally defined planning problem which can be checked by the planner (or a colouring problem which can checked by a program, etc), while in Jia et al. [2] there is a formal chemical composition and target property value that can be verified by the property checker.
>
> However, in our case, we start from an *informal* problem and the task is to arrive at the right formalization. Our proposer model proposes such formalizations as solutions, but *there does not exist any formal verifier that can check correctness* of a formalization with respect to an informal problem. Hence, the novelty in our approach is that the proposer model itself *also proposes the concrete instantiations* in addition to the formalization, and sends both of these to the verifier (the solver in our case). The verifier then only checks *consistency* between the concrete instances and the formalization rather than *correctness* of the formalization (as there is no formal way to check correctness in our case). We show how this notion of consistency yields near-perfect empirical precision in our setting. It is based on the concept that a proposer model is not likely to produce consistent abstract and concrete instances if it is not interpreting the problem correctly. In other prior settings where both proposer and verifier are informal predictors (e.g. LLMs self-critiquing), only relative improvements in accuracy are shown and our kind of near-perfect verification precision cannot be delivered – hence our novel ability to present verification as a standalone feature in this work.
>
> Our notion of consistency also ties into another angle of novelty in our technique, which is the new concept of *semantic ensembling* which we briefly discussed in the paper. The standard notion of ensembling in ML is that a combination/consensus of different predictors for a given task can yield improved accuracy. In semantic ensembling, we have predictors for *different* tasks (inferring concrete instantiations and abstract formalizations) whose outputs must satisfy mutual semantic satisfaction relationships that can be checked by an external solver, and we show how agreement in this kind of ensemble can yield near-perfect empirical precision. In summary, two key novel aspects of our technique are:
>
> - Our proposer model generates both the solution and the test cases while the verifier only checks consistency between these, as opposed to the standard paradigm where the verifier can check correctness of the solution with respect to the problem and produce counterexamples (since such correctness checking is not possible in our informal-to-formal prediction task).
> - Semantic ensembling is a new notion that combines predictors for *different* but semantically related tasks to achieve high precision, unlike standard ensembling which combines predictors for *the same* task.
>
> We hope that these novel conceptual aspects are effectively conveyed and we welcome further thoughts, questions or suggestions from the reviewers to articulate them better in the paper.

---

> ### Author Response · Authors · 2024-11-19
> **Response to Reviewer 9aEs [2/4]**
>
> **SSV’s success relies heavily on the LLM's ability to produce accurate formal representations and concrete instantiations**
>
> Yes, the SSV technique is parametric in the base LLM used and therefore its output depends on the underlying LLM quality. However, this is also true of the SoTA reasoning approaches which use LLMs or solver-augmented LLMs, and we show superior performance to these approaches using different base models. Another robustness aspect of SSV is its focus on precision by design:  as the quality of the abstract and concrete inferences reduces with weaker LLMs, they are also much less likely to agree with each other. This is empirically observed in our experiments with GPT3.5 where coverage fell significantly while precision was minimally affected.
>
> **Limited coverage raises questions about the method's practicality**
>
> In terms of practical impact, even such relatively low coverage can be directly beneficial in practice. Organizations commonly require very high (99%) accuracy if they are to deploy AI systems, especially moderate-to-high stakes systems (e.g. customer service chatbots, quality checks in manufacturing/logistics, document classification, etc).  If such an organization is presented with a system that has say 77% overall accuracy, then it may either not accept it or need to invest in manual human verification to check every output as there is no indication of when the system output is correct. However, if the system can provide verification with 99% precision on even 20% of cases, then that can reduce their overall cost of manual human verification by 20%. So this can have significant tangible impact in practice even if coverage is not that high.
>
> We would also like to emphasize that ours is the first work we know of that brings verification with such near-perfect precision to reasoning tasks for *any degree of coverage* – all previous works have made relative improvements to overall answer accuracy (which our method also delivers over SoTA). But providing almost perfect precision on a detectable subset of cases is an additional and novel contribution of this work. We have shown a significant coverage (ranging from 21.7% – 75.2% on the different datasets)  of detectable cases with such precision. We can also expect the coverage to increase with more powerful base LLM models as both the abstract and concrete instances can be expected to improve (as we saw with GPT3.5 vs. GPT4). We are looking forward to testing SSV with o1 as the base model once we get access/budget and testing even more complex benchmarks in other domains.
>
> **The paper could improve by providing a more detailed analysis of cases where SSV fails verification.**
>
> Thank you for this suggestion and it will indeed be a valuable addition to the paper. As checking the reasons for verification failures requires manual analysis, we re-executed the system and have logged and manually inspected a sample of 30 cases where verification fails. We have added a new section in the appendix (Appendix A.5) with the details of the failures and our analysis of the reasons, which the reviewer can find in the latest version of the paper. Here is a summary of the failure reasons:
>
> - code incorrect, example correct: 16 (53.3%)
> - code incorrect, example incorrect: 7 (23.3%)
> - code correct, example incorrect: 3 (10%)
> - program not well-formed: 4 (13.3%)
>
> We see that in most cases the code is incorrect as opposed to examples, which can be expected as examples inference is generally simpler than abstract translation. These results also highlight a central and novel principle of our approach that both examples and code inference can often be incorrect, but when they are consistent with each other then they are almost always correct, as shown by our near-perfect empirical precision of verification correctness (100% with GPT4 and 97% on GPT3.5).

---

> ### Author Response · Authors · 2024-11-19
> **Response to Reviewer 9aEs [3/4]**
>
> **Verifying complex or ambiguous natural language accurately.**
>
> Ambiguity/complexity of NL is a general problem for any reasoning approaches (and even for humans). In this work, our focus was not to specifically target this problem, but to show improvement over SoTA approaches in terms of accuracy and the ability to verify correctness with high precision on general NL-based reasoning problems, and we have shown this on the standard reasoning benchmarks. However, it will be interesting to explore how our approach behaves on very ambiguously presented NL tasks – we can expect that unlike other techniques, SSV may be able to detect such ambiguous cases through verification failures, since the abstract and concrete inferences are less likely to agree if the task is not clear. With such failures, we can imagine posing clarifying questions back to the user such as “is this an example of what you mean?” in an interactive manner. Much like how humans seek clarity through dialogue and examples when communicating in ambiguous NL.
>
> **How errors might occur if examples are not comprehensive.**
>
> While we have shown that even with two examples per constraint we can achieve full precision with GPT-4 on our benchmarks, in general one may generate an arbitrary number of examples to more comprehensively test different aspects of the constraint. There would be a trade-off in that more examples may increase verification precision but reduce coverage (since some examples may be incorrect), and this is something that may be experimented with and tweaked depending on application domains and preference for precision vs. coverage.
>
> **Runtime performance/computational costs**
>
> While our focus in this work was purely on accuracy and we have not yet implemented many obvious optimizations, we agree that runtime performance is an important aspect for practical application and have hence conducted an evaluation of the current system. Rerunning the current system over a sample of 250 data points (50 from each dataset), the median runtime per task is 152 seconds (around 2.5 minutes), with first quartile 108s, third quartile 267s and mean 249s. This was on an Intel Xeon Gold 6126 CPU @ 2.60 GHz with 16 cores and no hyper-threading, 62 GB of RAM, and an HDD-based storage system (machine has slightly lower single-threaded performance than modern desktops). However, there are many powerful optimizations that we can make to significantly reduce the run time:
>
> - The outer temperature loop (line 2 in Figure 4) can be fully parallelized as all the computations are independent for each temperature. That can yield up to 4X speed up (with 4 temperatures being tried in our current system). Side note: even with a single temperature of 0, our algorithm still beats all baselines in terms of accuracy (as in our ablation study), so even such an ablated system would be beneficial if computation costs are of significant concern.
> - In the verification phase (line 9 in Figure 4), the solver calls to verify each of the concrete instantiations can be parallelized as they are checked independently. These are around 10 to 20 independent solver calls on average (2 instantiations each for around 5-10 constraints) that can be parallelized for significant speedup.
> - Caching solver verification checks between repair attempts. Currently for each repair attempt in the inner loop (line 4 in Figure 4), we perform the full verification on the repaired program (on all constraints). However, most of the time the repaired change is on a single constraint for which a failing instantiation was found and all other constraints remain identical (though not always guaranteed as in some rare cases the LLM may reformulate the whole program). Hence if we cache the solver requests for each instantiation verification, many of these repetitive checks can be avoided in the repaired programs for the constraints that are unaltered.
>
> General side note: recent reasoning-oriented models such as Open AI’s o1 can take several seconds or up to a few minutes on some tasks (with significantly more computational resources/GPUs), so higher runtimes in the order of a few minutes with high computational resources may generally be expected to robustly address complex reasoning problems.

---

> ### Author Response · Authors · 2024-11-21
> **Response to Reviewer 9aEs [4/4]**
>
> **Large, unstructured data, and performance variations with different language models.**
>
> We have yet to explore application to any large unstructured data domains, but we can expect the reasoning aspects to often be orthogonal to the scale of data (which is usually homogenous in terms of logical properties). Performance variations with different LLMs: SSV offers a technique that can improve the reasoning power of any given LLM. So whichever LLM model is available to the user, SSV can provide an improved layer of robustness over it. While our core contribution is a robust system (based on the GPT4 LLM) that performs well above SoTA and provides reliable verification, generalizability to other LLMs is another  benefit the comes from the LLM-agnostic design of our SSV algorithm, and this will be an interesting dimension to explore more deeply in future work.
>
> **Its effectiveness outside formal logic tasks is still uncertain.**
>
> The focus of SSV is robustness of deductive logical reasoning, and does not apply in tasks that do not require such reasoning (e.g. information retrieval, sentiment analysis, poetry generation, etc). In a general purpose system (e.g. an open chatbot), one can imagine the system only invoking SSV if it detects a problem requiring sophisticated logical reasoning from the user (or if the response generated by the system involves such reasoning then it can be further verified with SSV). See also our discussion with reviewer “SW9F” on using SSV with augmented knowledge from LLMs to expand the scope beyond purely deductive inference tasks to reasoning with missing information.
>
> **Minor formatting issues:**
>
> Fixed, thank you.

---

> ### Author Response · Authors · 2024-11-25
>
> We have made further revisions to the paper (introduction section) to include the above points about our conceptual novelty in comparison with standard proposer-verifier approaches that use examples-based feedback.

---

### Official Review · Reviewer_SW9F · 2024-11-06

**Soundness:** 2
**Presentation:** 2
**Contribution:** 2
**Rating:** 6
**Confidence:** 3

**Summary:**

The authors propose SSV, a new reasoning approach to achieve "near-prefect" reasoning in LLMs using SMT solvers.
In their approach, the authors convert a query of text into a format readable by the Z3 theorem prover. They achieve this by converting each constraint from natural language (NL) to Z3 queries separately using an LLM. They then generate instances of the constraint in NL and also convert that to Z3 using an LLM. They use these as a sanity check to see whether the conversion was correct (although this is not guaranteed as pointed out by the authors). Once all of the constrainst are in place, the SMT solver will determine the solution.

**Strengths:**

The authors did a good job in keeping everything at a high level. From the writing and the examples, it is immidiately clear how they intend to mix LLMs with SMT solving using only high-level interfaces. The results also show that this approach has quite some potential, showing high precision. I also appreciate the attempt to incorporate SMT solvers into LLMs for better reasoning.

**Weaknesses:**

I would have liked to see more about performance. SMT solvers are known to be expensive in terms of required computational resources. Given that this approach uses the SMT solver in a nested loop, makes me question how scalable this is.

Furthermore, I am convinced that this will indeed work most of the time given that the entire context is given in text. However, if part of the reasoning is based on hidden but trivial facts, then the SMT solver will not know about these. I suspect that LLMs may generally be more accurate here, because they do have that context. E.g. how would this approach answer to this (or similar) query: "Urma is on earth. Urma releases a glass of water. What happens next? A. The glass falls. B The glass hovers in the air." GPT 3.5 is able to answer A and that the glass will fall because of gravity which is present due to Urma being on Earth, but the SMT solver does not have this context. I am not sure if this can be solved using the authors' approach. In that sense the author's approach is a natural language parser for an SMT solver that can attempt to self-repair the parsing when a counterexample is identified. I am not convinced that this approach contributes to "logical reasoning in LLMs" because one loses this context that is an essential feature of LLMs in my opinion.

**Questions:**

Specific comments:

Performance in terms of runtime

Work on the phrasing of your contribution. While this is a noteworthy paper, I think it should not be sold as contributing to logical reasoning in LLMs because I do not think that this holds for this approach. It is a more reliable parser from NL to SMT in my opinion.

Figure 2: What does it mean that the solver verification fails or succeeds for an initialization? To me it is unclear why one fails while the other does not. One or two lines on that would be appreciated.

Figure 4:   Line 6 states that we take the first answer to find as the best one, this does not make sense to me. I would say that an answer which has seen more counterexamples would be mopre trustworthy.
            Line 10 checks that P is well formed, why do we not check that before line 5?

257. "demarcated in explicit segments along with their NL descriptions (from the original problem) stated as comments." Add reference to Figure 1 for readability, at this point I had forgotten what constraints and options are.

348. Main results using GPT 4.0?

473. Related work: There are some papers explicitly on SMT solvers and LLMs (that I am not familiar with). I would have expected to see a comparison explicitly to these papers as well.

---

> ### Author Response · Authors · 2024-11-19
> **Response to Reviewer SW9F [1/3]**
>
> Thank you to the reviewer for the very helpful comments.
>
> **Runtime performance**
>
> While our focus in this work was purely on accuracy and we have not yet implemented many obvious optimizations, we agree that runtime performance is an important aspect for practical application and have hence conducted an evaluation of the current system. Rerunning the current system over a sample of 250 data points (50 from each dataset), the median runtime per task is 152 seconds (around 2.5 minutes), with first quartile 108s, third quartile 267s and mean 249s. This was on an Intel Xeon Gold 6126 CPU @ 2.60 GHz with 16 cores and no hyper-threading, 62 GB of RAM, and an HDD-based storage system (machine has slightly lower single-threaded performance than modern desktops).  However, there are many powerful optimizations that we can make to significantly reduce the run time:
> 1. The outer temperature loop (line 2 in Figure 4) can be fully parallelized as all the computations are independent for each temperature. That can yield up to 4X speed up (with 4 temperatures being tried in our current system). Side note: even with a single temperature of 0, our algorithm still beats all baselines in terms of accuracy (as in our ablation study), so even such an ablated system would be beneficial if computation costs are of significant concern.
> 2. In the verification phase (line 9 in Figure 4), the solver calls to verify each of the concrete instantiations can be parallelized as they are checked independently. These are around 10 to 20 independent solver calls on average (2 instantiations each for around 5-10 constraints) that can be parallelized for significant speedup.
> 3. Caching solver verification checks between repair attempts. Currently for each repair attempt in the inner loop (line 4 in Figure 4), we perform the full verification on the repaired program (on all constraints). However, most of the time the repaired change is on a single constraint for which a failing instantiation was found and all other constraints remain identical (though not always guaranteed as in some rare cases the LLM may reformulate the whole program). Hence if we cache the solver requests for each instantiation verification, many of these repetitive checks can be avoided in the repaired programs for the constraints that are unaltered.
>
> General side note: recent reasoning-oriented models such as Open AI’s o1 can take several seconds or up to a few minutes on some tasks (with significantly more computational resources/GPUs), so higher runtimes in the order of a few minutes may generally be expected to robustly address complex reasoning problems.

---

> > ### Comment · Reviewer_SW9F · 2024-11-19
> > **Comments**
> >
> > I want to thank the authors for doing these performance tests. I think these results are important to mention, especially when using SMT-solvers. The optimizations seem reasonable as well, although the SMT solver itself will remain a bottleneck. However, that I am willing to accept.
> >
> > However, I would like the authors to still comment on how to deal with facts that are obvious from context (see weaknesses).
> >
> > I would also like to see the authors' view on the alternative phrasing of the contribution that I provided.

---

> > > ### Author Response · Authors · 2024-11-25
> > >
> > > Thank you for your kind response. We wanted to confirm that we have responded to all of your other comments. We have also made the further revisions to the paper to include the runtime performance results in the evaluation section (Section 4.1) and in more detail in Appendix A.6, as well as a brief discussion of reasoning with missing knowledge in the limitations section (Section 5).

---

> ### Author Response · Authors · 2024-11-19
> **Response to Reviewer SW9F [2/3]**
>
> **Reasoning with hidden information**
>
> You make a very valid point about hidden information and it also raises some very interesting prospects.
>
> For your specific example of Urma’s glass, we tested it directly with our SSV technique and it does produce the correct answer A. The reason for this is that the logical formulation indeed was not able to infer either option A or B (because of the missing information as you point out) and hence the algorithm resorted to direct LLM inference (line 14 in our algorithm). This shows that the SSV approach is robust to missing information in the sense that it falls back to LLM inference when an answer cannot be inferred with logical inference (this fallback is possible since the problem of missing information is logically detectable). However, this does not address your core point that pure logical inference cannot handle such tasks directly, and this raises very interesting prospects.
>
> LLMs and solvers have complementary strengths. As you say solvers require all the information in the premises as they are doing strict logical inference, while LLMs can reason with very likely implicit/hidden background knowledge. On the other hand, solvers can guarantee correct deductive inference while LLMs can make mistakes in this. So we can define two subproblems to address for reasoning in general:
>
> 1. Deduction: When given all required premises, can we infer a logically correct solution?
> 2. Background knowledge inference: if some information is missing or not explicitly stated, can we infer this?
>
> In this work our focus has been on problem (1) for valid deduction, which is a challenging open problem that we have addressed with significant improvements and robustness over SoTA (and we will clarify this scope of focus in the paper). However, to expand the scope to include reasoning with  hidden/likely background knowledge, we may further utilize LLMs to address problem (2).  Given a question, we can first ask the LLM to explicitly add any hidden background information that may be required for logical inference, and then send this augmented question to our SSV technique. This way, when the final answer is obtained, the system can also inform the user of the additional assumptions it made in its reasoning to arrive at that answer (in case the user is not happy with the assumptions). We made the following simple one-shot prompt to test this:
>
> ```
> Can you change the given question to add any simple explicit constraints about background knowledge that would be required for a logical solver to logically deduce the answer.
> Question:
> It is night time. Which is true? A. It is dark B. It is bright
> UpdatedQuestion.
> It is night time. At night it is dark. Which is true? A. It is dark B. It is bright
> Question:
> Urma is on earth. Urma releases a glass of water. What happens next? A. The glass falls. B The glass hovers in the air.
> UpdatedQuestion:
> ```
> and sending this to GPT4 we get this augmented question that includes the missing information about gravity causing objects to fall:
>
> ```
> Urma is on Earth, where gravity causes objects to fall when released. Urma releases a glass of water. What happens next? A. The glass falls. B. The glass hovers in the air.
> ```
>
> When we send this question to SSV, then it is able to infer a solver program to get the correct answer (A), as it now explicitly receives the constraint that “gravity causes objects to fall when released”. This is a very interesting prospect to incorporate the common sense knowledge and abductive/inductive inference of LLMs with the pure deductive inference of logical solvers! Another similar approach could be to ask the LLM to first produce its reasoning to answer the problem, and then use SSV to check the LLM’s reasoning steps formally and iterate over this if necessary. This would be a great direction for next steps to expand the scope of solver-based techniques beyond pure deductive reasoning.

---

> ### Author Response · Authors · 2024-11-19
> **Response to Reviewer SW9F [3/3]**
>
> **Phrasing of contribution**
>
> Yes we are indeed addressing the formalization problem and we can clarify this positioning further in the paper. However, we do believe it is not just a parsing problem but requires reasoning and modelling to correctly formulate the problem in the solver from the high level NL descriptions.  Since the solver is doing the automated deductive reasoning, the key missing step in end-to-end reasoning from natural language is the formalization step. This is the main problem with solver-augmented LLM approaches in general (Pan et al. (2023), Ye et al. (2023)), and the problem of Autoformalization is the central challenge in many formal reasoning domains such as planning (Kambhampati et al. (2024); Guan et al. (2024)), formal proofs (Wu et al. (2022); Jiang et al. (2023); He-Yueya et al. (2023)). We are addressing it here for general logical reasoning tasks. However, it is not just a matter of parsing the natural language to solver code: a key challenge is that the natural language is at a high level of abstraction and so the system must *formulate* or model the problem correctly in the low level solver language, e.g. choosing appropriate concrete data structures/types  and expressing the different high level constraints posed in the problem using these (and there can be many such alternative formulations). This problem formulation aspect requires reasoning in itself – which our SSV technique addresses with the combination of abstract and concrete inferences to refine and verify the formulations to ensure robustness. For instance, if a question mentions some events that occur in time order then a simple solver formulation may just assign numbers to events to indicate their position, but if a constraint mentions that some event happens in the “afternoon” then this may fail our concrete instantiation tests, and the system would need to reformulate the problem to include more explicit temporal information. So we would say that we are addressing the deeper formulation problem rather than just parsing.
>
> **Figure 2: What does it mean that the solver verification fails or succeeds for an initialization?**
>
> We will clarify this. Formally, if we have a constraint C and a positive example (instantiation) I in solver code, then we add both C and I as constraints to the solver. We then query the solver to check if there is any satisfying model for these constraints. If the constraints are not satisfiable then that means the instantiation I is not possible under the constraint C, and otherwise it is permitted. If we have a negative example instantiation N, then we add both C and N to the solver and check unsatisfiability: if they are  unsatisfiable then N is a valid negative example.
>
> **Figure 4: Line 6 states that we take the first answer to find as the best one ... I would say that an answer which has seen more counterexamples would be more trustworthy.**
>
> We will clarify this step in the paper. In the repair attempts loop the goal is to stop repairs as soon as a program that produces some answer is found (since beyond this, further repairs may force an agreement between the examples and program that introduces bias in the verification). So we only use the examples-based refinement until we can find some valid program. What we should clarify is that the loop stopping actually comes from the RepairProgram() function at line 12: if P produces some answer then this function does not repair it and just returns ∅, so that the repair loop is terminated as soon as some A_best is found.
>
> **Line 10 checks that P is well formed, why do we not check that before line 5?**
>
> At line 5, if P is well-formed, we must still perform verification. If it is not well-formed, we still perform verification as that can help infer failing instantiations to improve the program. So in both cases it does not help to check well-formedness at this point.
>
> **"demarcated in explicit segments along with their NL descriptions (from the original problem) stated as comments." Add reference to Figure 1 for readability**
>
> Fixed, thank you.
>
> **Main results using GPT 4.0?**
>
> We used GPT 4v version 2024-02-15-preview (from Azure Open AI API).
>
> **Related work: There are some papers explicitly on SMT solvers and LLMs (that I am not familiar with). I would have expected to see a comparison explicitly to these papers as well.**
>
> The two closest works using LLMs with SMT solvers for logical reasoning tasks that we know of are Pan et al. (2023) and Ye et al. (2023), where Pan et al show the SoTA results especially on the most difficult AR-LSAT dataset. That is why we compared to this technique on all the same datasets and showed significant improvement in accuracy and the additional ability to verify correctness with 100% precision over a significant coverage of cases.

---

> > ### Comment · Reviewer_SW9F · 2024-11-25
> > **RE: Phrasing of contribution**
> >
> > I agree that you have a contribution that is slightly more than a conventional NL parser as I noted in my review. The problem that I am having is the fact that none of the reasoning of the SMT solver is invisible to the LLM. In other words, it converts the NL to SMT and then outputs the results of the SMT solver (or if that fails it generates one using the LLM). However, this approach does not allow me to ask why that is the answer. This is because the SSV is not doing any reasoning, it is just bluntly throwing models at an SMT solver hoping for an answer.
> >
> > I agree that this research is worthwhile, but I think that you cannot say that it "advances the robustness of reasoning with large language models" (as per your conclusion) as the LLM parses the NL to SMT (using your technique which is the actual contribution) and then leverages the actual reasoning to the SMT solver.

---

> ### Author Response · Authors · 2024-11-26
>
> Thank you for your very valuable further feedback and we hope to achieve more clarity on this important point of phrasing that you raise.
>
> **On this claim: "advances the robustness of reasoning with LLMs"**
>
> At a high level, the overall reasoning problem is this: given a deductive reasoning question in natural language can we find the right answer. Any AI system that uses LLMs to address this problem can be described as a system that is "reasoning with LLMs". Our SSV system is one such system (irrespective of how it is implemented) and we show it to have increased robustness over other such SoTA systems. So this was the intent of the claim that it "advances the robustness of reasoning with LLMs". However, we agree that the phrasing is vague and there can be misinterpretations, so we can revise this by saying "advances the robustness of AI reasoning systems by inferring verified problem formalizations through a novel combination of LLMs and logical solvers."
>
> As an analogy, for data analytics tasks, a technique that uses LLMs to formulate the correct SQL query from an NL query much better than SoTA can be said to "advance the robustness of data analytics with LLMs", even though the actual analysis of the data is being done by the SQL engine rather than the technique itself or the LLM. Similarly, in our case the *formal* part of reasoning inference can be done by the SMT engine, but the challenging part of constructing *the right* SMT query is done by our SSV technique (using the novel combination of LLMs and the solver itself).
>
> **"this approach does not allow me to ask why that is the answer"**
>
> This is a good point. While our scope in this work was only to address the final answer accuracy, it is certainly possible to obtain the detailed proofs from the SMT solver (e.g. setting the proof generation option to true in the Z3 solver). This provides the detailed steps of reasoning done by the solver. This can either be shown directly to the user or also paraphrased in natural language using LLMs for better readability.
>
>
> **"none of the reasoning of the SMT solver is visible to the LLM"**
>
> This is true and is part of the design of any such tool-augmented LLM approaches: that the *formal* part of reasoning can be factored out and offloaded to a solver (that guarantees correctness of for this part) and the *informal* part of reasoning is done with help of the LLM. This is why such approaches are often referred to as "LLM-modulo" approaches (e.g. Kambhampati et al, ICML 2024). The formal reasoning happening inside the solver is not needed to be visible to the LLM as the LLM is only used to infer the correct formalization of the problem in the first place (if a formalization is incorrect to begin with then the formal reasoning that the solver does on it is not useful). This gives a clean separation of concerns between the formal and the informal part of the overall reasoning task.
>
>
> **"SSV is not doing any reasoning, it is just bluntly throwing models at an SMT solver hoping for an answer"**
>
> We would say that SSV is doing significant work on the *informal* part of reasoning that is vital to converge on the correct formalization of the problem. It uses many notions of reasoning, such as decomposing a complex problem in NL into sub-problems (inferring sub-constraints), to infer concrete instantiations to check the correctness of a formalization for each constraint (verification), to refine the formalization with respect to concrete instantiations (repair loop), to try different alternative formulations of the problem (different temperatures). These are all notions of reasoning often used by humans when approaching a complex task. As an analogy: consider a human needing to write code to solve a problem. They would first decompose the problem into sub-problems that can be implemented separately (our sub-constraints), think of unit tests to check the correctness of components (our concrete instantiations), and if they fail then they refine the components (our repair loop), and if an entire formulation is not working then they may consider alternative formulations (our temperature based exploration). All of this is done to arrive at the correct formalization which is the final code. So while the final code can be executed to solve the problem formally and is doing all the formal problem solving, the process of arriving at it required many notions of iterative reasoning involving the human and the code execution engine to ensure that the correct code is written (which hopefully is considered more than just a parser from NL to the final code). Similarly, SSV implements all of these notions of reasoning combining the LLM and the solver to arrive at the right solver code.
>
> We hope that these points provide more clarity and that our alternative phrasing is acceptable, and if there are more concerns we would be happy to address them.

---

> ### Comment · Reviewer_SW9F · 2024-11-27
> **Final verdict**
>
> I want to thank the authors for their response. I have taken into account the comments and the other reviews.
>
> I do think that the idea is good, but I agree with the other authors that the contribution is small.
>
> I also still disagree with the presentation in the paper. I think the contribution is reliable translation of NL into an SMT model and nothing more. Evidence for this is provided by the authors. In my opinion though, this approach is still far away from logical reasoning in LLMs and I think that the authors focus too much on that aspect in the introduction and conclusion.
>
> I therefore think my confusion may be a presentation issue and I updated the score accordingly. My final verdict is that I cannot increase the total rating any further.

---

> > ### Author Response · Authors · 2024-11-27
> >
> > Thank you to the reviewer for your very valuable feedback and discussion, and we really appreciate your update to the ratings. We have revised the paper further to rephrase the conclusion and parts of the introduction to further clarify that our work focusses on advancing reasoning by addressing the key challenge of formalization and correctly formulating the problem from NL to the solver code.

---

### Author Response · Authors · 2024-12-01

We sincerely thank the reviewers for the very valuable discussion. It has greatly helped to emphasize the distinctive contribution of this work in many important dimensions:

- **Unprecedented Verification Feature.** We have introduced a novel verification feature where the system guarantees answer correctness with near-perfect certainty (100% correctness on standard benchmarks). No prior work in natural language reasoning has provided such a capability (to the best of our knowledge).

- **Strong Empirical Advancement.** Apart from the near-certain verification, we have significantly advanced answer accuracy on reasoning tasks above SoTA approaches consistently over standard reasoning benchmarks, and in particular from 43% to 71% on the most challenging law school tests benchmark.

- **Core Conceptual Novelty.** In contrast to the standard proposer-verifier paradigm where formal verifiers can be used to check *correctness* in formal problem task domains (which is inherently impossible in our informal-to-formal setting), we show how one can leverage *consistency* between abstract and concrete inferences to use formal verifiers for informal-to-formal prediction tasks. This novel consistency-based paradigm provides near-certain correctness where full correctness is impossible.

Apart from the significant concrete advancements, these novel contributions establish a new research direction: aiming for near-certain verification in natural language reasoning using formal methods, which is an orthogonal and complementary direction to the ongoing focus on only improving answer accuracy. We hope that the foundational nature of these contributions and their potential to inspire future research can be recognized.

---

### Public Comment · ~Mohammad_Raza1 · 2025-02-09
**Misrepresentation in the meta-review summary**

Thank you again to all the reviewers for their time and feedback. To avoid potential confusions, we want to clarify that there is a significant misrepresentation of this work in the meta-review summary:

*"This paper combines LLMs with logical solvers, where the LLM is set to present a formal representation of a problem to a (SMT) solver. The core motivation is to shift the responsibility of the full reasoning from the LLM to the solver."*

This describes exactly the existing baseline works (e.g., Pan et al. EMNLP’23, Ye et al. NeurIPS’23) and does not reflect the new contribution of this work. This paper instead addresses a key limitation of such LLM+solver approaches: that the LLM very often generates the wrong problem formalization. It proposes a novel method that leverages solver-based consistency between abstract and concrete inferences by the LLM to address this issue.

---

### Meta-Review · Area_Chair_VPDh · 2024-12-19

**Metareview:**

This paper combines LLMs with logical solvers, where the LLM is set to present a formal representation of a problem to a (SMT) solver. The core motivation is to shift the responsibility of the full reasoning from the LLM to the solver.

While the reviewers generally agree that the paper provides a good idea, all reviewers also agree that the contribution is rather small and in the current state of the paper presented too boldly. The authors are encouraged to sharpen the message to the actual technical contribution. In the current form, the paper cannot be accepted.

**Additional Comments On Reviewer Discussion:**

The extensive answers of the authors were appreciated, but could not dissolve the concerns of the reviewers.

---

### Decision · Program_Chairs · 2025-01-22

Reject